# INPUT-GRADIENT SPACE PARTICLE INFERENCE FOR NEURAL NETWORK ENSEMBLES

**Trung Trinh**[1]     **Markus Heinonen**[1]     **Luigi Acerbi**[2]     **Samuel Kaski**[1,3]

[1]Department of Computer Science, Aalto University, Finland
[2]Department of Computer Science, University of Helsinki, Finland
[3]Department of Computer Science, University of Manchester, United Kingdom
{trung.trinh, markus.o.heinonen, samuel.kaski}@aalto.fi,
luigi.acerbi@helsinki.fi

## ABSTRACT

Deep Ensembles (DEs) demonstrate improved accuracy, calibration and robustness to perturbations over single neural networks partly due to their functional diversity. Particle-based variational inference (ParVI) methods enhance diversity by formalizing a repulsion term based on a network similarity kernel. However, weight-space repulsion is inefficient due to over-parameterization, while direct function-space repulsion has been found to produce little improvement over DEs. To sidestep these difficulties, we propose First-order Repulsive Deep Ensemble (FoRDE), an ensemble learning method based on ParVI, which performs repulsion in the space of first-order input gradients. As input gradients uniquely characterize a function up to translation and are much smaller in dimension than the weights, this method guarantees that ensemble members are functionally different. Intuitively, diversifying the input gradients encourages each network to learn different features, which is expected to improve the robustness of an ensemble. Experiments on image classification datasets and transfer learning tasks show that FoRDE significantly outperforms the gold-standard DEs and other ensemble methods in accuracy and calibration under covariate shift due to input perturbations.

## 1 INTRODUCTION

Ensemble methods, which combine predictions from multiple models, are a well-known strategy in machine learning (Dietterich, 2000) to boost predictive performance (Lakshminarayanan et al., 2017), uncertainty estimation (Ovadia et al., 2019), robustness to adversarial attacks (Pang et al., 2019) and corruptions (Hendrycks & Dietterich, 2019). Deep ensembles (DEs) combine multiple neural networks from independent weight initializations (Lakshminarayanan et al., 2017). While DEs are simple to implement and have promising performance, their weight-based diversity does not necessarily translate into useful functional diversity (Rame & Cord, 2021; D'Angelo & Fortuin, 2021; Yashima et al., 2022).

Particle-based variational inference (ParVI) (Liu & Wang, 2016; Chen et al., 2018; Liu et al., 2019; Shen et al., 2021) has recently emerged as a direction to promote diversity in neural ensembles from the Bayesian perspective (Wang et al., 2019; D'Angelo & Fortuin, 2021). Notably, the ParVI update rule adds a kernelized repulsion term $k(f, f')$ between the ensemble networks $f, f'$ for explicit control of the ensemble diversity. Typically repulsion is done in the weight space to capture different regions in the weight posterior. However, due to the over-parameterization of neural networks, weight-space repulsion suffers from redundancy. An alternative approach is to define the repulsion in function space (Wang et al., 2019; D'Angelo & Fortuin, 2021), which requires the challenging computation of a kernel between functions. Previous works avoided this issue by comparing functions only on training inputs, which leads to underfitting (D'Angelo et al., 2021). Neither weight nor function space repulsion has led to significant improvements over vanilla DEs.

From a functional perspective, a model can *also* be uniquely represented, up to translation, using its first-order derivatives, i.e., input *gradients* $\nabla_{\mathbf{x}} f$. Promoting diversity in this third view of input gradients has notable advantages:

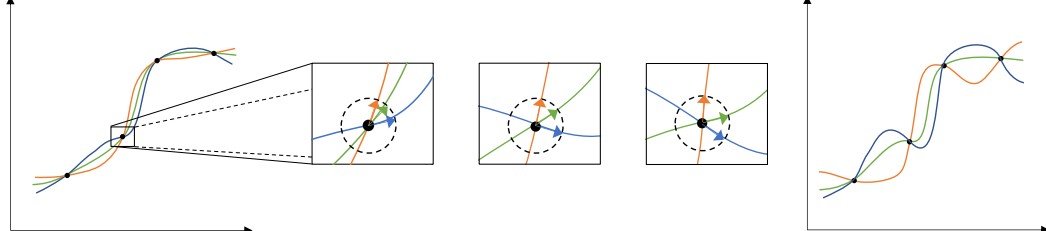

Figure 1: **Input-gradient repulsion increases functional diversity.** An illustration of input gradient repulsion in 1D regression with 3 neural networks. **Left:** At some point during training, the models fit well to the training samples yet exhibit low functional diversity. **Middle**: As training proceeds, at each data point, the repulsion term gradually pushes the input gradients (represented by the arrows) away from each other on a unit sphere. **Right:** As a result, at the end of training, the ensemble has gained functional diversity.

1. each ensemble member is guaranteed to correspond to a different function;

2. input gradients have smaller dimensions than weights and thus are more amenable to kernel comparisons;

3. unlike function-space repulsion, input-gradient repulsion does not lead to training point underfitting (See Fig. 1 and the last panel of Fig. 2);

4. each ensemble member is encouraged to learn different features, which can improve robustness.

In this work, we propose a ParVI neural network ensemble that promotes diversity in their input gradients, called First-order Repulsive deep ensemble (FoRDE). Furthermore, we devise a data-dependent kernel that allows FoRDE to outperform other ensemble methods under input corruptions on image classification tasks. Our code is available at https://github.com/AaltoPML/FoRDE.

## 2 BACKGROUND

**Bayesian neural networks** In a Bayesian neural network (BNN), we treat the model's weights $\theta$ as random variables with a prior $p(\theta)$. Given a dataset $\mathcal{D} = \{(\mathbf{x}_n, y_n)\}_{n=1}^N$ and a likelihood function $p(y|\mathbf{x}, \theta)$ per data point, we infer the posterior over weights $p(\theta|\mathcal{D})$ using Bayes' rule

$$p(\theta|\mathcal{D}) = \frac{p(\mathcal{D}|\theta)p(\theta)}{\int_\theta p(\mathcal{D}|\theta)p(\theta)\mathrm{d}\theta} = \frac{p(\theta)\prod_{n=1}^N p(y_n|\mathbf{x}_n, \theta)}{\int_\theta p(\theta)\prod_{n=1}^N p(y_n|\mathbf{x}_n, \theta)\mathrm{d}\theta}, \tag{1}$$

where the likelihood is assumed to factorize over data. To make a prediction on a test sample $\mathbf{x}^*$, we integrate over the inferred posterior in Eq. (1), a practice called *Bayesian model averaging* (BMA):

$$p(y|\mathbf{x}^*, \mathcal{D}) = \int_\theta p(y|\mathbf{x}^*, \theta)p(\theta|\mathcal{D})\mathrm{d}\theta = \mathbb{E}_{p(\theta|\mathcal{D})}\big[p(y|\mathbf{x}^*, \theta)\big]. \tag{2}$$

However, computing the integral in Eq. (2) is intractable for BNNs. Various approximate inference methods have been developed for BNNs, including variational inference (VI) (Graves, 2011; Blundell et al., 2015), Markov chain Monte Carlo (MCMC) (Neal, 2012; Welling & Teh, 2011; Zhang et al., 2020) and more recently ParVI (Liu & Wang, 2016; Wang et al., 2019; D'Angelo & Fortuin, 2021).

**Deep ensembles** As opposed to BNNs, which attempt to learn the posterior distribution, DEs (Lakshminarayanan et al., 2017) consist of multiple maximum-a-posteriori (MAP) estimates trained from independent random initializations. They can capture diverse functions that explain the data well, as independent training runs under different random conditions will likely converge to different modes in the posterior landscape. DEs have been shown to be better than BNNs in both accuracy and uncertainty estimation (Ovadia et al., 2019; Ashukha et al., 2020; Gustafsson et al., 2020).

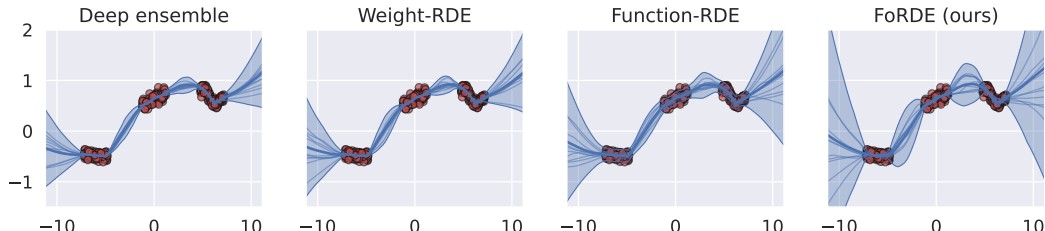

Figure 2: **Input gradient ensembles (FoRDE) capture higher uncertainty than baselines.** Each panel shows predictive uncertainty in 1D regression for different (repulsive) deep ensemble methods.

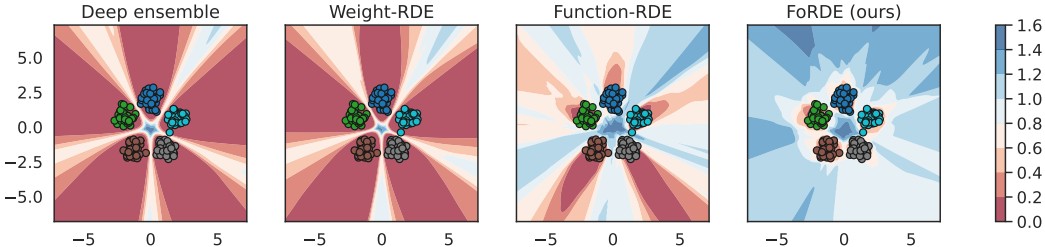

Figure 3: **Uncertainty of FoRDE is high in all input regions outside the training data, and is higher than baselines.** Each panel shows the entropy of the predictive posteriors in 2D classification.

**Particle-based variational inference for neural network ensembles**  ParVI methods (Liu & Wang, 2016; Chen et al., 2018; Liu et al., 2019; Shen et al., 2021) have been studied recently to formalize neural network ensembles. They approximate the target posterior using a set of samples, or particles, by deterministically transporting these particles to the target distribution (Liu & Wang, 2016). ParVI methods are expected to be more efficient than MCMC as they take into account the interactions between particles in their update rules (Liu et al., 2019). These repulsive interactions are driven by a kernel which measures the pairwise similarities between particles, i.e., networks (Liu et al., 2019).

The current approaches compare networks in weight space $\theta$ or in function space $f(\cdot; \theta)$. Weight-space repulsion is ineffective due to difficulties in comparing extremely high-dimensional weight vectors and the existence of weight symmetries (Fort et al., 2019; Entezari et al., 2022). Previous studies show that weight-space ParVI does not improve performance over plain DEs (D'Angelo & Fortuin, 2021; Yashima et al., 2022). Comparing neural networks via a function kernel is also challenging since functions are infinite-dimensional objects. Previous works resort to comparing functions only on a subset of the input space (Wang et al., 2019; D'Angelo & Fortuin, 2021). Comparing functions over training data leads to underfitting (D'Angelo & Fortuin, 2021; Yashima et al., 2022), likely because these inputs have known labels, leaving no room for diverse predictions without impairing performance.

## 3 FORDE: FIRST-ORDER REPULSIVE DEEP ENSEMBLES

In this section, we present a framework to perform ParVI in the *input-gradient* space. We start by summarizing Wasserstein gradient descent (WGD) in Section 3.1, and show how to apply WGD for input-gradient-space repulsion in Section 3.2. We then discuss hyperparameter selection for the input-gradient kernel in Section 3.3, and outline practical considerations in Section 3.4.

Throughout this paper, we assume a set of $M$ weight particles $\{\theta_i\}_{i=1}^{M}$ corresponding to a set of $M$ neural networks $\{f_i : \mathbf{x} \mapsto f(\mathbf{x}; \theta_i)\}_{i=1}^{M}$. We focus on the supervised classification setting: given a labelled dataset $\mathcal{D} = \{(\mathbf{x}_n, y_n)\}_{n=1}^{N}$ with $\mathcal{C}$ classes and inputs $\mathbf{x}_n \in \mathbb{R}^D$, we approximate the posterior $p(\theta|\mathcal{D})$ using the $M$ particles. The output $f(\mathbf{x}; \theta)$ for input $\mathbf{x}$ is a vector of size $\mathcal{C}$ whose $y$-th entry $f(\mathbf{x}; \theta)_y$ is the logit of the $y$-th class.

## 3.1 Wasserstein gradient descent

Following D'Angelo & Fortuin (2021), we use a ParVI method called Wasserstein gradient descent (WGD) (Liu et al., 2019; Wang et al., 2022). Given an intractable target posterior distribution $\pi = p(\cdot|\mathcal{D})$ and a set of particles $\{\theta_i\}_{i=1}^M$ from distribution $\rho$, the goal of WGD is to find the particle distribution $\rho^*$,

$$\rho^* = \underset{\rho \in \mathcal{P}_2(\Theta)}{\arg\min} \, \mathrm{KL}_\pi(\rho), \tag{3}$$

where $\mathrm{KL}_\pi(\rho)$ is a shorthand for the standard Kullback-Leibler divergence

$$\mathrm{KL}_\pi(\rho) = \mathbb{E}_{\rho(\theta)}\big[\log \rho(\theta) - \log \pi(\theta)\big], \tag{4}$$

and $\mathcal{P}_2(\Theta)$ is the Wasserstein space equipped with the Wasserstein distance $W_2$ (Ambrosio et al., 2005; Villani, 2009). WGD solves the problem in Eq. (3) using a Wasserstein gradient flow $(\rho_t)_t$, which is roughly the family of steepest descent curves of $\mathrm{KL}_\pi(\cdot)$ in $\mathcal{P}_2(\Theta)$. The tangent vector of this gradient flow at time $t$ is

$$v_t(\theta) = \nabla \log \pi(\theta) - \nabla \log \rho_t(\theta) \tag{5}$$

whenever $\rho_t$ is absolutely continuous (Villani, 2009; Liu et al., 2019). Intuitively, $v_t(\theta)$ points to the direction where the probability mass at $\theta$ of $\rho_t$ should be transported in order to bring $\rho_t$ closer to $\pi$. Since Eq. (5) requires the analytical form of $\rho_t$ which we do not have access to, we use kernel density estimation (KDE) to obtain a tractable approximation $\hat{\rho}_t$ induced by particles $\{\theta_i^{(t)}\}_{i=1}^M$ at time $t$,

$$\hat{\rho}_t(\theta) \propto \sum_{i=1}^M k\big(\theta, \theta_i^{(t)}\big), \tag{6}$$

where $k$ is a positive semi-definite kernel. Then, the gradient of the approximation is

$$\nabla \log \hat{\rho}_t(\theta) = \frac{\sum_{i=1}^M \nabla_\theta k\big(\theta, \theta_i^{(t)}\big)}{\sum_{i=1}^M k\big(\theta, \theta_i^{(t)}\big)}. \tag{7}$$

Using Eq. (7) in Eq. (5), we obtain a practical update rule for each particle $\theta^{(t)}$ of $\hat{\rho}_t$:

$$\theta^{(t+1)} = \theta^{(t)} + \eta_t \left( \underbrace{\nabla_{\theta^{(t)}} \log \pi\big(\theta^{(t)}\big)}_{\text{driving force}} - \underbrace{\frac{\sum_{i=1}^M \nabla_{\theta^{(t)}} k\big(\theta^{(t)}, \theta_i^{(t)}\big)}{\sum_{i=1}^M k\big(\theta^{(t)}, \theta_i^{(t)}\big)}}_{\text{repulsion force}} \right), \tag{8}$$

where $\eta_t > 0$ is the step size at optimization time $t$. Intuitively, we can interpret the first term in the particle gradient as the driving force directing the particles towards high density regions of the posterior, while the second term is the repulsion force pushing the particles away from each other.

## 3.2 Defining the kernel for WGD in input gradient space

We propose to use a kernel comparing the *input gradients* of the particles,

$$k(\theta_i, \theta_j) \overset{\text{def}}{=} \mathbb{E}_{(\mathbf{x},y) \sim p(\mathbf{x},y)}\Big[\kappa\big(\nabla_\mathbf{x} f(\mathbf{x}; \theta_i)_y, \nabla_\mathbf{x} f(\mathbf{x}; \theta_j)_y\big)\Big], \tag{9}$$

where $\kappa$ is a *base kernel* between gradients $\nabla_\mathbf{x} f(\mathbf{x}; \theta)_y$ that are of same size as the inputs $\mathbf{x}$. In essence, we define $k$ as the expected similarity between the input gradients of two networks with respect to the data distribution $p(\mathbf{x}, y)$. Interestingly, by using the kernel $k$, the KDE approximation $\hat{\rho}$ of the particle distribution not only depends on the particles themselves but also depends on the data distribution. We approximate the kernel $k$ using the training samples, with linear complexity:

$$k(\theta_i, \theta_j) \approx k_\mathcal{D}(\theta_i, \theta_j) = \frac{1}{N} \sum_{n=1}^N \kappa\big(\nabla_\mathbf{x} f(\mathbf{x}_n; \theta_i)_{y_n}, \nabla_\mathbf{x} f(\mathbf{x}_n; \theta_j)_{y_n}\big). \tag{10}$$

The kernel only compares the gradients of the true label $\nabla_\mathbf{x} f(\mathbf{x}_n; \theta)_{y_n}$, as opposed to the entire Jacobian matrix $\nabla_\mathbf{x} f(\mathbf{x}_n; \theta)$, as our motivation is to encourage each particle to learn different features that could explain the training sample $(\mathbf{x}_n, y_n)$ well. This approach also reduces computational complexity, since automatic differentiation libraries such as JAX (Bradbury et al., 2018) or Pytorch (Paszke et al., 2019) would require $\mathcal{C}$ passes, one per class, to calculate the full Jacobian.

**Choosing the base kernel**  We choose the RBF kernel on the unit sphere as our base kernel $\kappa$:

$$\kappa(\mathbf{s}, \mathbf{s}'; \boldsymbol{\Sigma}) = \exp\left(-\frac{1}{2}(\mathbf{s} - \mathbf{s}')^\top \boldsymbol{\Sigma}^{-1}(\mathbf{s} - \mathbf{s}')\right), \qquad \mathbf{s} = \frac{\nabla_\mathbf{x} f(\mathbf{x}; \theta)_y}{||\nabla_\mathbf{x} f(\mathbf{x}; \theta)_y||_2} \in \mathbb{R}^D \quad (11)$$

where $\mathbf{s}, \mathbf{s}'$ denote the two normalized gradients of two particles with respect to one input, and $\boldsymbol{\Sigma} \in \mathbb{R}^{D \times D}$ is a diagonal matrix containing squared lengthscales. We design $\kappa$ to be norm-agnostic since the norm of the true label gradient $||\nabla_\mathbf{x} f(\mathbf{x}_n; \theta)_{y_n}||_2$ fluctuates during training and as training converges, the log-probability $f(\mathbf{x}_n; \theta)_{y_n} = \log p(y_n|\mathbf{x}_n, \theta)$ will approach $\log 1$, leading to the norm $||\nabla_\mathbf{x} f(\mathbf{x}_n; \theta)_{y_n}||_2$ approaching 0 due to the saturation of the log-softmax activation. Furthermore, comparing the normed input gradients between ensemble members teaches them to learn complementary explanatory patterns from the training samples, which could improve robustness of the ensemble. The RBF kernel is an apt kernel to compare unit vectors (Jayasumana et al., 2014), and we can control the variances of the gradients along input dimensions via the square lengthscales $\boldsymbol{\Sigma}$.

### 3.3 SELECTING THE LENGTHSCALES FOR THE BASE KERNEL

In this section, we present a method to select the lengthscales for the base kernel. These lengthscales are important for the performance of FoRDE, since they control how much repulsion force is applied in each dimension of the input-gradient space. The dimension-wise repulsion is (Liu & Wang, 2016)

$$\frac{\partial}{\partial s_d}\kappa(\mathbf{s}, \mathbf{s}'; \boldsymbol{\Sigma}) = -\frac{s_d - s'_d}{\boldsymbol{\Sigma}_{dd}}\kappa(\mathbf{s}, \mathbf{s}'; \boldsymbol{\Sigma}), \quad (12)$$

where we can see that along the $d$-th dimension the inverse square lengthscale $\boldsymbol{\Sigma}_{dd}$ controls the strength of the repulsion $\nabla_{s_d}\kappa(\mathbf{s}, \mathbf{s}'; \boldsymbol{\Sigma})$: a smaller lengthscale corresponds to a stronger force.[1] Additionally, since the repulsion is restricted to the unit sphere in the input-gradient space, increasing distance in one dimension decreases the distance in other dimensions. As a result, the repulsion motivates the ensemble members to depend more on dimensions with stronger repulsion in the input gradient space for their predictions, while focusing less on dimensions with weaker repulsion. One should then apply stronger repulsion in dimensions of data manifold with higher variances.

To realize the intuition above in FoRDE, we first apply Principal Component Analysis (PCA) to discover dominant features in the data. In PCA, we calculate the eigendecomposition

$$\mathbf{C} = \mathbf{U}\boldsymbol{\Lambda}\mathbf{U}^T \quad \in \mathbb{R}^{D \times D} \quad (13)$$

of the covariance matrix $\mathbf{C} = \mathbf{X}^T\mathbf{X}/(N-1)$ of the centered training samples $\mathbf{X} \in \mathbb{R}^{N \times D}$ to get eigenvectors and eigenvalues $\{\mathbf{u}_d, \lambda_d\}_{k=1}^D$. The $d$-th eigenvalue $\lambda_d$ is the variance of the data along eigenvector $\mathbf{u}_d$, offering a natural choice of inverse eigenvalues $\lambda_d^{-1}$ as the squared lengthscales $\boldsymbol{\Sigma}_{dd}$ of the principal components. Let $\tilde{\mathbf{x}} = \mathbf{U}^T\mathbf{x}$ denote the representation of the input $\mathbf{x}$ in eigenbasis $\mathbf{U} = [\mathbf{u}_1 \, \mathbf{u}_2 \, \ldots \, \mathbf{u}_D]$. We compute the gradient kernel in PCA basis $\mathbf{U}$ and set $\boldsymbol{\Sigma} = \boldsymbol{\Lambda}^{-1}$:

$$\kappa(\mathbf{s}, \mathbf{s}') \stackrel{\text{def}}{=} \exp\left(-\frac{1}{2}(\mathbf{U}^\top\mathbf{s} - \mathbf{U}^\top\mathbf{s}')^\top \boldsymbol{\Lambda}(\mathbf{U}^\top\mathbf{s} - \mathbf{U}^\top\mathbf{s}')\right), \quad (14)$$

where $\boldsymbol{\Lambda}$ is a diagonal eigenvalue matrix. While setting the square inverse lengthscales equal to the eigenvalues seems problematic at first glance, since large eigenvalues will push the kernel $\kappa$ towards 0, this problem is avoided in practice since we also employ the median heuristic, which introduces in the kernel a global bandwidth scaling term that adapts to the current pairwise distance between particles, as discussed below in Section 3.4.

**Connection to the EmpCov prior**  Recently, Izmailov et al. (2021a) proposed the EmpCov prior for the weight columns $\mathbf{w}$ of the first layer of a BNN:

$$\mathbf{w} \sim \mathcal{N}(0, \alpha\mathbf{C} + \epsilon\mathbf{I}), \qquad \mathbf{w} \in \mathbb{R}^D \quad (15)$$

where $\alpha > 0$ determines the prior scale and $\epsilon > 0$ is a small constant ensuring a positive definite covariance. The prior encourages first layer weights to vary more along higher variance data dimensions. Samples from this prior will have large input gradients along high variance input dimensions. In this sense, the EmpCov prior has a similar effect to the kernel in Eq. (14) on ensemble members. The difference is that while Izmailov et al. (2021a) incorporates knowledge of the data manifold into the prior, we embed this knowledge into our approximate posterior via the kernel.

---

[1]Here we assume that the lengthscales are set appropriately so that the kernel $\kappa$ does not vanish, which is true since we use the median heuristic during training (Section 3.4).

### 3.4 PRACTICAL CONSIDERATIONS

In this section, we detail two important considerations to make FoRDEs work in practice. We include the full training algorithm in Appendix C.1.

**Mini-batching**  To make FoRDE amenable to mini-batch gradient optimization, we adapt the kernel in Eq. (10) to a mini-batch of samples $\mathcal{B} = \{(\mathbf{x}_b, y_b)\}_{b=1}^B$:

$$k_{\mathcal{B}}(\theta_i, \theta_j) = \frac{1}{B} \sum_{b=1}^B \kappa \Big( \nabla_{\mathbf{x}} f(\mathbf{x}_b; \theta_i)_{y_b}, \nabla_{\mathbf{x}} f(\mathbf{x}_b; \theta_j)_{y_b} \Big). \tag{16}$$

In principle, this kernel in the update rule in Eq. (8) leads to biased stochastic gradients of the repulsion term because the average over batch samples in Eq. (16) is inside the logarithm. However, in practice, we found no convergence issues in our experiments.

**Median heuristics**  Since we perform particle optimization with an RBF kernel $\kappa$, following earlier works (Liu & Wang, 2016; Liu et al., 2019), we adopt the median heuristic (Schölkopf et al., 2002). Besides the lengthscales, we introduce a global bandwidth $h$ in our base kernel in Eq. (11):

$$\kappa(\mathbf{s}_i, \mathbf{s}_j; \mathbf{\Sigma}) = \exp \left( -\frac{1}{2h}(\mathbf{s}_i - \mathbf{s}_j)^\top \mathbf{\Sigma}^{-1}(\mathbf{s}_i - \mathbf{s}_j) \right), \qquad \mathbf{s}_i = \frac{\nabla_{\mathbf{x}} f(\mathbf{x}; \theta_i)_y}{||\nabla_{\mathbf{x}} f(\mathbf{x}; \theta_i)_y||_2} \in \mathbb{R}^D. \tag{17}$$

During training, the bandwidth $h$ is adaptively set to $\mathrm{med}^2/(2 \log M)$, where $\mathrm{med}^2$ is the median of the pairwise distance $(\mathbf{s}_i - \mathbf{s}_j)^\top \mathbf{\Sigma}^{-1}(\mathbf{s}_i - \mathbf{s}_j)$ between the weight samples $\{\theta_i\}_{i=1}^M$.

### 3.5 COMPUTATIONAL COMPLEXITY

Compared to DEs, FoRDEs take roughly three times longer to train. In addition to a forward-backward pass to calculate the log-likelihood, we need an additional forward-backward pass to calculate the input gradients, and another backward pass to calculate the gradients of the input gradient repulsion with respect to the weights. This analysis is confirmed in practice: in RESNET18/CIFAR-100 experiments of Section 5.2 with an ensemble size of 10, a DE took ∼31 seconds per epoch on an Nvidia A100 GPU, while FoRDE took ∼101 seconds per epoch.

## 4 RELATED WORKS

Besides the ParVI methods mentioned in Section 2, we discuss additional related works below.

**Diversifying input gradients of ensembles**  Local independent training (LIT) (Ross et al., 2018) orthogonalizes the input gradients of ensemble members by minimizing their pairwise squared cosine similarities, and thus closely resembles our input-gradient repulsion term which diversifies input gradients on a hyper-sphere. However, their goal is to find a maximal set of models that accurately predict the data using different sets of distinct features, while our goal is to induce functional diversity in an ensemble. Furthermore, we formulate our kernelized repulsion term based on the ParVI framework, allowing us to choose hyperparameter settings (orthogonal basis and lengthscales) that imbue the ensemble with beneficial biases (such as robustness to corruption).

**Gradient-based attribution methods for deep models**  One application of input gradients is to build attribution (or *saliency*) maps, which assign importance to visually-interpretable input features for a specified output (Simonyan et al., 2014; Bach et al., 2015; Shrikumar et al., 2016; Sundararajan et al., 2017; Shrikumar et al., 2017). Our method intuitively utilizes the attribution perspective of input gradients to encourage ensemble members to learn different patterns from training data.

**Improving corruption robustness of BNNs**  Previous works have evaluated the predictive uncertainty of BNNs under covariate shift (Ovadia et al., 2019; Izmailov et al., 2021b), with Izmailov et al. (2021b) showing that standard BNNs with high-fidelity posteriors perform worse than MAP solutions on under corruptions. Izmailov et al. (2021a) attributed this phenomenon to the lack of posterior contraction in the null space of the data manifold and proposed the EmpCov prior as a

remedy. As discussed in Section 3.3, the PCA kernel works in the same manner as the EmpCov prior and thus significantly improves robustness of FoRDE against corruptions. Trinh et al. (2022) studied the robustness of node-BNNs, an efficient alternative to weight-based BNNs, and showed that by increasing the entropy of the posterior, node-BNNs become more robust against corruptions. Wang & Aitchison (2023) allow BNNs to adapt to the distribution shift at test time by using test data statistics.

Table 1: **FoRDE-PCA achieves the best performance under corruptions while FoRDE-Identity outperforms baselines on clean data. FoRDE-Tuned outperforms baselines on both clean and corrupted data.** Results of RESNET18 / CIFAR-100 averaged over 5 seeds. Each ensemble has 10 members. cA, cNLL and cECE are accuracy, NLL, and ECE on CIFAR-100-C.

| METHOD | NLL ↓ | ACCURACY (%) ↑ | ECE ↓ | cA / cNLL / cECE |
|---|---|---|---|---|
| NODE-BNNS | 0.74±0.01 | 79.7±0.3 | 0.054±0.002 | 54.8 / 1.96 / 0.05 |
| SWAG | 0.73±0.01 | 79.4±0.1 | **0.038±0.001** | 53.0 / 2.03 / 0.05 |
| DEEP ENSEMBLES | **0.70±0.00** | 81.8±0.2 | 0.041±0.003 | 54.3 / 1.99 / 0.05 |
| WEIGHT-RDE | **0.70±0.01** | 81.7±0.3 | 0.043±0.004 | 54.2 / 2.01 / 0.06 |
| FUNCTION-RDE | 0.76±0.02 | 80.1±0.4 | 0.042±0.005 | 51.9 / 2.08 / 0.07 |
| FEATURE-RDE | 0.75±0.04 | **82.1±0.3** | 0.072±0.023 | 54.8 / 2.02 / 0.06 |
| LIT | **0.70±0.00** | 81.9±0.2 | 0.040±0.003 | 54.4 / 1.98 / 0.05 |
| FoRDE-PCA (OURS) | 0.71±0.00 | 81.4±0.2 | 0.039±0.002 | **56.1 / 1.90 / 0.05** |
| FoRDE-IDENTITY (OURS) | **0.70±0.00** | **82.1±0.2** | 0.043±0.001 | 54.1 / 2.02 / 0.05 |
| FoRDE-TUNED (OURS) | **0.70±0.00** | **82.1±0.2** | 0.044±0.002 | 55.3 / 1.94 / 0.05 |

Table 2: **FoRDE-PCA achieves the best performance under corruptions while FoRDE-Identity has the best NLL on clean data. FoRDE-Tuned outperforms most baselines on both clean and corrupted data.** Results of RESNET18 / CIFAR-10 averaged over 5 seeds. Each ensemble has 10 members. cA, cNLL and cECE are accuracy, NLL, and ECE on CIFAR-10-C.

| METHOD | NLL ↓ | ACCURACY (%) ↑ | ECE ↓ | cA / cNLL / cECE |
|---|---|---|---|---|
| NODE-BNNS | 0.127±0.009 | 95.9±0.3 | 0.006±0.002 | 78.2 / 0.82 / 0.09 |
| SWAG | 0.124±0.001 | **96.9±0.1** | 0.005±0.001 | 77.5 / 0.78 / 0.07 |
| DEEP ENSEMBLES | 0.117±0.001 | 96.3±0.1 | 0.005±0.001 | 78.1 / 0.78 / 0.08 |
| WEIGHT-RDE | 0.117±0.002 | 96.2±0.1 | 0.005±0.001 | 78.0 / 0.78 / 0.08 |
| FUNCTION-RDE | 0.128±0.001 | 95.8±0.2 | 0.006±0.001 | 77.1 / 0.81 / 0.08 |
| FEATURE-RDE | 0.116±0.001 | 96.4±0.1 | **0.004±0.001** | 78.1 / 0.77 / 0.08 |
| LIT | 0.116±0.001 | 96.4±0.1 | 0.004±0.001 | 78.2 / 0.78 / 0.09 |
| FoRDE-PCA (OURS) | 0.125±0.001 | 96.1±0.1 | 0.006±0.001 | **80.5 / 0.71 / 0.07** |
| FoRDE-IDENTITY (OURS) | **0.113±0.002** | 96.3±0.1 | 0.005±0.001 | 78.0 / 0.80 / 0.08 |
| FoRDE-TUNED (OURS) | 0.114±0.002 | 96.4±0.1 | 0.005±0.001 | 79.1 / 0.74 / 0.07 |

Table 3: **FoRDE outperforms EmpCov priors under corruptions, while maintaining competitive performance on clean data.** Results of RESNET18 on CIFAR-10 evaluated over 5 seeds. Each ensemble has 10 members. cA, cNLL and cECE are accuracy, NLL, and ECE on CIFAR-10-C. Here we use the EmpCov prior for all methods except FoRDE.

| METHOD | NLL ↓ | ACCURACY (%) ↑ | ECE ↓ | cA / cNLL / cECE |
|---|---|---|---|---|
| DEEP ENSEMBLES | 0.119±0.001 | **96.2±0.1** | 0.006±0.001 | 78.7 / 0.76 / 0.08 |
| WEIGHT-RDE | 0.120±0.001 | 96.0±0.1 | 0.006±0.001 | 78.8 / 0.76 / 0.08 |
| FUNCTION-RDE | 0.132±0.001 | 95.6±0.3 | 0.007±0.001 | 77.8 / 0.79 / 0.08 |
| FEATURE-RDE | **0.118±0.001** | **96.2±0.1** | **0.005±0.001** | 78.9 / 0.74 / 0.07 |
| FoRDE-PCA (OURS) | 0.125±0.001 | 96.1±0.1 | 0.006±0.001 | **80.5 / 0.71 / 0.07** |

## 5 EXPERIMENTS

### 5.1 ILLUSTRATING FUNCTIONAL DIVERSITY

To show that FoRDE does produce better functional diversity than plain DE and other repulsive DE approaches, we repeated the 1D regression of Izmailov et al. (2019) and the 2D classification

experiments of D'Angelo & Fortuin (2021). We use ensembles of 16 networks for these experiments. Fig. 2 shows that FoRDE exhibits higher predictive uncertainty in the input regions outside the training data compared to the baselines in 1D regression. For the 2D classification task, we visualize the entropy of the predictive posteriors in Fig. 3, which also shows that FoRDE has higher uncertainty than the baselines. Furthermore, FoRDE is the only method that exhibits high uncertainty in all input regions outside the training data, a property mainly observed in predictive uncertainty of Gaussian processes (Rasmussen & Williams, 2006).

## 5.2 COMPARISONS TO OTHER REPULSIVE DE METHODS AND BNNs

We report performance of FoRDE against other methods on CIFAR-10/100 (Krizhevsky, 2009) in Tables 1–2 and TINYIMAGENET (Le & Yang, 2015) in Appendix A. Besides the PCA lengthscales introduced in Section 3.3, we experiment with the identity lengthscales $\Sigma = I$ and with tuned lengthscales where we take the weighted average of the PCA lengthscales and the identity lengthscales. Details on lengthscale tuning are presented in Appendix D.4. For the repulsive DE (RDE) baselines, we choose weight RDE (D'Angelo & Fortuin, 2021), function RDE (D'Angelo & Fortuin, 2021) and feature RDE (Yashima et al., 2022). We also include LIT (Ross et al., 2018), node-BNNs (Trinh et al., 2022) and SWAG (Maddox et al., 2019) as baselines. We use an ensemble size of 10. We use standard performance metrics of expected calibration error (ECE) (Naeini et al., 2015), negative log-likelihood (NLL) and predictive accuracy. For evaluations on input perturbations, we use CIFAR-10/100-C and TINYIMAGENET-C provided by Hendrycks & Gimpel (2017), which are datasets of corrupted test images containing 19 image corruption types across 5 levels of severity, and we report the accuracy, NLL and ECE averaged over all corruption types and severity levels (denoted cA, cNLL and cECE in Tables 1–4). We use RESNET18 (He et al., 2016a) for CIFAR-10/100 and PREACTRESNET18 (He et al., 2016b) for TINYIMAGENET. Experimental details are included in Appendix C.2.

Tables 1 and 2 show that FoRDE-PCA outperforms other methods under input corruptions across all metrics, while maintaining competitive performance on clean data. For instance, FoRDE-PCA shows a $+1.3\%$ gain on CIFAR-100-C and $+2.4\%$ gain on CIFAR-10-C in accuracy compared to the second-best results. As stated in Section 3.3, the PCA kernel encourages FoRDE to rely more on features with high variances in the data manifold to make predictions, while being less dependent on features with low variances. This effect has been shown in Izmailov et al. (2021a) to boost model robustness against perturbations, which explains why FoRDE with the PCA kernel performs better than the baselines on input corruptions.

On the other hand, Tables 1 and 2 show that FoRDE with identity lengthscales outperforms the baselines in terms of NLL on CIFAR-10 and has the best accuracy on CIFAR-100. However, FoRDE-Identity is slightly less robust than DE against corruptions. We suspect that with the identity lengthscales, FoRDE also learns to rely on low-variance features to make predictions, which is harmful to performance under corruptions (Izmailov et al., 2021a).

Finally, Tables 1 and 2 show that FoRDE with tuned lengthscales (FoRDE-Tuned) outperforms the baselines on both clean and corrupted data in most cases, suggesting that the optimal lengthscales for good performance on both clean and corrupted data lie somewhere between the identity lengthscales and the PCA lengthscales. Additional results on lengthscale tuning are presented in Appendix D.4.

## 5.3 COMPARISONS TO EMPCOV PRIOR

As stated in Section 3.3, our approach is similar to the EmpCov prior (Izmailov et al., 2021a). We thus perform comparisons against ensemble methods where the EmpCov prior is defined for the first layer instead of the standard isotropic Gaussian prior. We report the results of RESNET18/CIFAR-10 in Table 3, where we use the EmpCov prior for all ensemble methods except FoRDE. Comparing Table 3 to Table 2 indicates that the EmpCov prior slightly improves robustness of the baseline ensemble methods against corruptions, while also leading to a small reduction in performance on clean data. These small improvements in robustness are not surprising, since for ensemble methods consisting of approximate MAP solutions, the isotropic Gaussian prior already minimizes the influences of low-variance data features on the ensemble's predictions (Izmailov et al., 2021a). We argue that besides minimizing the influences of low variance features on predictions via the PCA kernel, FoRDE also encourages its members to learn complementary patterns that can explain the data well, and these

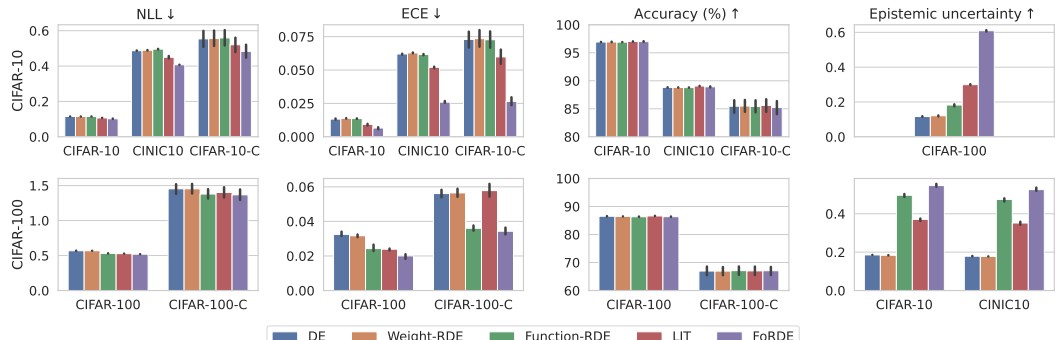

Figure 4: **FoRDE outperforms competing methods in transfer learning. First three columns:** We report NLL, ECE and accuracy on in-distribution test sets and under covariate shift. For CIFAR-10, we use CINIC10 (Darlow et al., 2018) to evaluate models under natural shift and CIFAR-10-C for corruption shift. For CIFAR-100, we evaluate on CIFAR-100-C. FoRDE performs better than the baselines in all cases. **Last column:** We evaluate functional diversity by calculating epistemic uncertainty of ensembles on out-of-distribution (OOD) datasets using the formula in Depeweg et al. (2018). We use CIFAR-100 as the OOD test set for CIFAR-10 and we use CIFAR-10 and CINIC10 as OOD test sets for CIFAR-100. FoRDE exhibits higher functional diversity than the baselines.

two effects act in synergy to improve the robustness of the resulting ensemble. Thus, Table 3 shows that FoRDE is still more robust against corruptions than the baseline methods with the EmpCov prior.

## 5.4 TRANSFER LEARNING EXPERIMENTS

To show the practicality of FoRDE, we evaluated its performance in a transfer learning scenario. We use the outputs of the last hidden layer of a Vision Transformer model pretrained on IMAGENET-21K as input features and train ensembles of 10 networks. We report the results on CIFAR-10 in the first row and on CIFAR-100 in the second row in Fig. 4. Overall, Fig. 4 shows that FoRDE is better than the baselines across all cases. See Appendix C.3 for experimental details.

## 6 DISCUSSION

In this section, we outline directions to further improve FoRDE.

**Reducing computational complexity** One major drawback of FoRDEs is the high computational complexity as discussed in Section 3.5. To circumvent this problem, one could either (i) only calculate the repulsion term after every $k > 1$ epochs, or (ii) using only a subset of batch samples at each iteration to calculate the repulsion term.

**Reducing the number of lengthscale parameters** Here we use the RBF kernel as our base kernel, which requires us to choose appropriate lengthscales for good performance. To avoid this problem, we could explore other kernels suitable for unit vector comparisons, such as those introduced in Jayasumana et al. (2014). Another solution is to study dimensionality reduction techniques for input gradients before calculating the kernel, which can reduce the number of lengthscales to be set.

## 7 CONCLUSION

In this work, we proposed FoRDE, an ensemble learning method that promotes diversity in the input-gradient space among ensemble members. We detailed the update rule and devised a data-dependent kernel suitable for input-gradient repulsion. Experiments on image classification and transfer learning tasks show that FoRDE outperforms other ensemble methods under covariate shift. Future directions include more efficient implementations of the method and reducing the burden of hyperparameter selection as discussed in Section 6.

ACKNOWLEDGMENTS

This work was supported by the Research Council of Finland (Flagship programme: Finnish Center for Artificial Intelligence FCAI and decision no. 359567, 345604 and 341763), ELISE Networks of Excellence Centres (EU Horizon: 2020 grant agreement 951847) and UKRI Turing AI World-Leading Researcher Fellowship (EP/W002973/1). We acknowledge the computational resources provided by Aalto Science-IT project and CSC–IT Center for Science, Finland.

ETHICS STATEMENT

Our paper introduces a new ensemble learning method for neural networks, allowing deep learning models to be more reliable in practice. Therefore, we believe that our work contributes towards making neural networks safer and more reliable to use in real-world applications, especially those that are safety-critical. Our technique per se does not directly deal with issues such as fairness, bias or other potentially harmful societal impacts, which may be caused by improper usages of machine learning or deep learning systems (Mehrabi et al., 2021). These issues would need to be adequately considered when constructing the datasets and designing specific deep learning applications.

REPRODUCIBILITY STATEMENT

For the purpose of reproducibility of our results with our new ensemble learning method, we have included in the Appendix detailed descriptions of the training algorithm. For each experiment, we include in the Appendix details about the neural network architecture, datasets, data augmentation procedures and hyperparameter settings. All datasets used for our experiments are publicly available. We have included our codes in the supplementary material and we provide instructions on how to run our experiments in a `README.md` available in the provided codebase. For the transfer learning experiments, we used publicly available pretrained models which we have mentioned in the Appendix.

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

## A    RESULTS ON TINYIMAGENET

Table 4: **FoRDE-PCA performs best under corruptions while having competitive performance on clean data.** Results of PREACTRESNET18 on TINYIMAGENET evaluated over 5 seeds. Each ensemble has 10 members. cA, cNLL and cECE are accuracy, NLL, and ECE on TINYIMAGENET-C.

| METHOD | NLL ↓ | ACCURACY (%) ↑ | ECE ↓ | cA / cNLL / cECE |
|---|---|---|---|---|
| NODE-BNNS | 1.39±0.01 | 67.6±0.3 | 0.114±0.004 | 30.4 / 3.40 / **0.05** |
| SWAG | 1.39±0.01 | 66.6±0.3 | **0.020**±**0.005** | 28.4 / 3.72 / 0.11 |
| DEEP ENSEMBLES | **1.15**±**0.00** | 71.6±0.0 | 0.035±0.002 | 31.8 / 3.38 / 0.09 |
| WEIGHT-RDE | **1.15**±0.01 | 71.5±0.0 | 0.036±0.003 | 31.7 / 3.39 / 0.09 |
| FUNCTION-RDE | 1.21±0.02 | 70.2±0.5 | 0.036±0.004 | 31.1 / 3.43 / 0.10 |
| FEATURE-RDE | 1.24±0.01 | **72.0**±**0.1** | 0.100±0.003 | 31.9 / 3.35 / 0.09 |
| LIT | **1.15**±**0.00** | 71.5±0.0 | 0.035±0.002 | 31.2 / 3.40 / 0.11 |
| FoRDE-PCA (OURS) | 1.16±0.00 | 71.4±0.0 | 0.033±0.002 | **32.2** / **3.28** / 0.08 |

## B    PERFORMANCE UNDER DIFFERENT ENSEMBLE SIZES

We report the NLL of FoRDE and DE under different ensemble sizes on CIFAR-10/100 and CIFAR-10/100-C in Figs. 5–6. We use the WIDERESNET16X4 (Zagoruyko & Komodakis, 2016) architecture for this experiment. These figures show that both methods enjoy significant improvements in performance as the ensemble size increases. While Fig. 5a and Fig. 6a show that FoRDE underperforms DE on clean images, Fig. 5b and Fig. 6b show that FoRDE significantly outperforms DE on corrupted images, such that a FoRDE with 10 members has the same or better corruption robustness of a DE with 30 members.

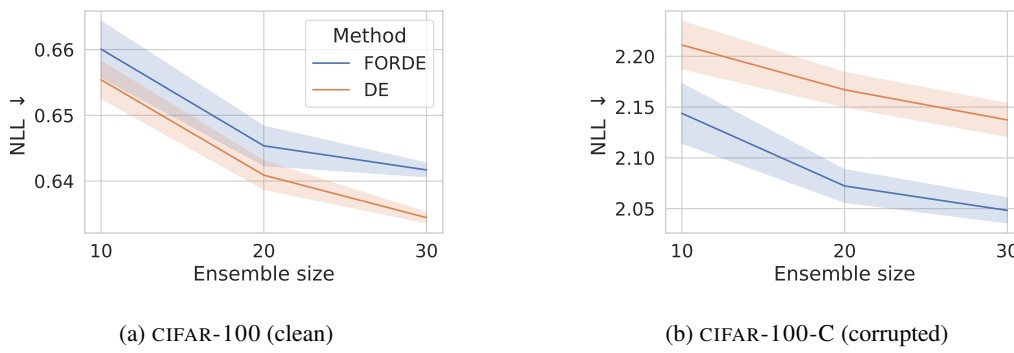

(a) CIFAR-100 (clean)                    (b) CIFAR-100-C (corrupted)

Figure 5: **FoRDE is competitive on in-distribution and outperforms DEs under domain shifts by corruption**. Performance of WIDERESNET16X4 on CIFAR-100 over 5 seeds.

## C    TRAINING PROCEDURE

### C.1    TRAINING ALGORITHM FOR FoRDE

We describe the training algorithm of FoRDE in Algorithm 1.

### C.2    EXPERIMENTAL DETAILS FOR IMAGE CLASSIFICATION EXPERIMENTS

For all the experiments, we used SGD with Nesterov momentum as our optimizer, and we set the momemtum coefficient to 0.9. We used a weight decay $\lambda$ of $5 \times 10^{-4}$ and we set the learning rate $\eta$ to $10^{-1}$. We used a batch size of 128 and we set $\epsilon$ in Algorithm 1 to $10^{-12}$. We used 15 bins to calculate ECE during evaluation.

---

**Algorithm 1** FoRDE

---

1: **Input:** training data $\mathcal{D}$, orthonormal basis $\mathbf{U}$, diagonal matrix of squared lengthscales $\boldsymbol{\Sigma}$, a neural network ensemble $\{f(\cdot;\theta_i)\}_{i=1}^M$ of size $M$, positive scalar $\epsilon$, number of iterations $T$, step sizes $\{\eta_t\}_{t=1}^T$, weight decay $\lambda$

2: **Output:** optimized parameters $\{\theta_i^{(T)}\}_{i=1}^M$

3: Initialize parameters $\{\theta_i^{(0)}\}_{i=1}^M$

4: **for** $t = 1$ **to** $T$ **do**

5:      Draw a mini-batch $\{\mathbf{x}_b, y_b\}_{b=1}^B \sim \mathcal{D}$.

6:      **for** $b = 1$ **to** $B$ **do**

7:          **for** $i = 1$ **to** $M$ **do**      ▷ Calculate the normalized input gradients for each $\theta_i$ (Eq. (11))

8:

$$\mathbf{s}_{i,b} \longleftarrow \frac{\nabla_{\mathbf{x}_b} f\big(\mathbf{x}_b; \theta_i^{(t)}\big)_{y_b}}{\sqrt{\big|\big|\nabla_{\mathbf{x}_b} f\big(\mathbf{x}_b; \theta_i^{(t)}\big)_{y_b}\big|\big|_2^2 + \epsilon^2}} \tag{18}$$

9:          **end for**

10:          **for** $i = 1$ **to** $M$ **do**      ▷ Calculate the pairwise squared distance in Eq. (14)

11:             **for** $j = 1$ **to** $M$ **do**

$$d_{i,j,b} \longleftarrow \frac{1}{2}(\mathbf{s}_{i,b} - \mathbf{s}_{j,b})^\top \mathbf{U}\boldsymbol{\Sigma}\mathbf{U}^\top(\mathbf{s}_{i,b} - \mathbf{s}_{j,b}) \tag{19}$$

12:             **end for**

13:          **end for**

14:          Calculate the global bandwidth per batch sample using the median heuristic (Eq. (17)):

$$h_b \longleftarrow \text{median}\big(\{d_{i,j,b}\}_{i=1,j=1}^{M,M}\big)/(2\ln M) \tag{20}$$

15:      **end for**

16:      **for** $i = 1$ **to** $M$ **do**      ▷ Calculate the pairwise kernel similarity using Eq. (16) and Eq. (17)

17:          **for** $j = 1$ **to** $M$ **do**

$$k_{i,j} \longleftarrow \frac{1}{B}\sum_{b=1}^B \exp\big(-d_{i,j,b}/h_b\big) \tag{21}$$

18:          **end for**

19:      **end for**

20:      **for** $i = 1$ **to** $M$ **do**

21:          Calculate the gradient of the repulsion term using Eq. (7):

$$\mathbf{g}_i^{\text{rep}} \longleftarrow \frac{\sum_{j=1}^M \nabla_{\theta_i^{(t)}} k_{i,j}}{\sum_{j=1}^M k_{i,j}} \tag{22}$$

22:          Calculate the gradient $\mathbf{g}_i^{\text{data}}$ of the cross-entropy loss with respect to $\theta_i$.

23:          Calculate the update vector in Eq. (8):

$$\mathbf{v}_i^{(t)} \longleftarrow \frac{1}{B}\big(\mathbf{g}_i^{\text{data}} - \mathbf{g}_i^{\text{rep}}\big) \tag{23}$$

24:          Update the parameters and apply weight decay:

$$\theta_i^{(t+1)} \longleftarrow \theta_i^{(t)} + \eta_t(\mathbf{v}_i^{(t)} - \lambda\theta_i^{(t)}) \tag{24}$$

25:      **end for**

26: **end for**

---

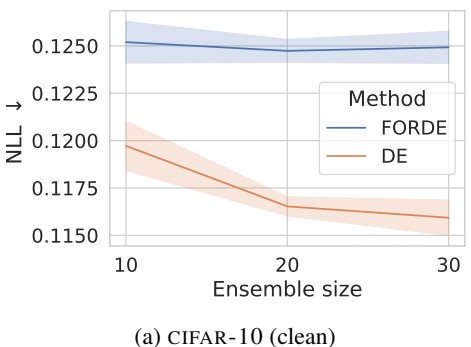
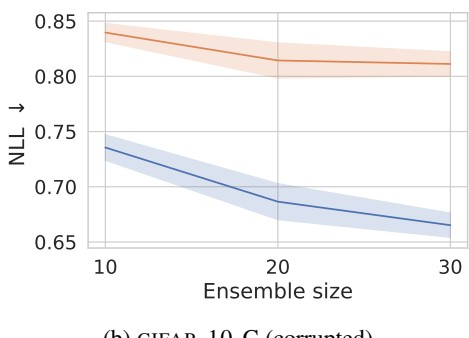

(a) CIFAR-10 (clean)                    (b) CIFAR-10-C (corrupted)

Figure 6: **FoRDE is competitive on in-distribution and outperforms DEs under domain shifts by corruption**. Performance of WIDERESNET16X4 on CIFAR-10 over 5 seeds.

On CIFAR-10 and CIFAR-100, we use the standard data augmentation procedure, which includes input normalization, random cropping and random horizontal flipping. We ran each experiments for 300 epochs. We decreased the learning rate $\eta$ linearly from $10^{-1}$ to $10^{-3}$ from epoch 150 to epoch 270. For evaluation, we used all available types for corruptions and all levels of severity in CIFAR-10/100-C.

On TINYIMAGENET, we use the standard data augmentation procedure, which includes input normalization, random cropping and random horizontal flipping. We ran each experiments for 150 epochs. We decreased the learning rate $\eta$ linearly from $10^{-1}$ to $10^{-3}$ from epoch 75 to epoch 135. For evaluation, we used all available types for corruptions and all levels of severity in TINYIMAGENET-C.

For weight-RDE and FoRDE, we only imposed a prior on the weights via the weight decay parameter. For feature-RDE and function-RDE, we followed the recommended priors in Yashima et al. (2022). For feature-RDE, we used Cauchy prior with a prior scale of $10^{-3}$ for CIFAR-10 and a prior scale of $5 \times 10^{-3}$ for both CIFAR-100 and TINYIMAGENET, and we used a projection dimension of 5. For function-RDE, we used Cauchy prior with a prior scale of $10^{-6}$ for all datasets.

### C.3   EXPERIMENTAL DETAILS FOR TRANSFER LEARNING EXPERIMENTS

We extracted the outputs of the last hidden layer of a Vision Transformer model pretrained on the ImageNet-21k dataset (`google/vit-base-patch16-224-in21k` checkpoint in the `transformers` package from `huggingface`) and use them as input features, and we trained ensembles of 10 ReLU networks with 3 hidden layers and batch normalization.

For all the experiments, we used SGD with Nesterov momentum as our optimizer, and we set the momemtum coefficient to 0.9. We used a batch size of 256, and we annealed the learning rate from 0.2 to 0.002 during training. We used a weight decay of $5 \times 10^{-4}$. We used 15 bins to calculate ECE for evaluation. For OOD experiments, we calculated epistemic uncertainty on the test sets of CIFAR-10/100 and CINIC10. For evaluation on natural corruptions, we used all available types for corruptions and all levels of severity in CIFAR-10/100-C.

## D   ADDITIONAL RESULTS

### D.1   INPUT GRADIENT DIVERSITY AND FUNCTIONAL DIVERSITY

To show that FoRDE indeed produces ensembles with higher input gradient diversity among member models, which in turn leads to higher functional diversity than DE, we visualize the input gradient distance and epistemic uncertainty of FoRDE and DE in Fig. 7. To measure the differences between input gradients, we use *cosine distance*, defined as $1 - \cos(\mathbf{u}, \mathbf{v})$ where $\cos(\mathbf{u}, \mathbf{v})$ is the cosine similarity between two vectors $\mathbf{u}$ and $\mathbf{v}$. To quantify functional diversity, we calculate the epistemic uncertainty using the formula in Depeweg et al. (2018), similar to the transfer learning experiments. Fig. 7 shows that FoRDE has higher gradient distances among members compared to DE, while also having higher epistemic uncertainty across all levels of corruption severity. Intuitively, as the test

inputs become more corrupted, epistemic uncertainty of both FoRDE and DE increases, and the input gradients between member models become more dissimilar for both methods. These results suggest that there could be a connection between input gradient diversity and functional diversity in neural network ensembles.

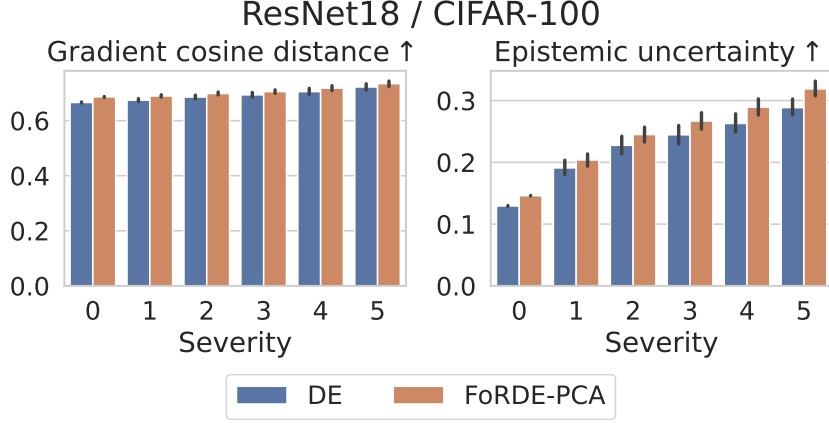

Figure 7: **FoRDE has higher gradient distance as well as higher epistemic uncertainty** Results of RESNET18 on CIFAR100 over 5 seeds under different levels of corruption severity, where 0 mean no corruption.

## D.2 PERFORMANCE UNDER CORRUPTIONS

We plot performance of all methods under the RESNET18/CIFAR-C setting in Figs. 8 and 9. These figures show that FoRDE achieves the best performance across all metrics under all corruption severities.

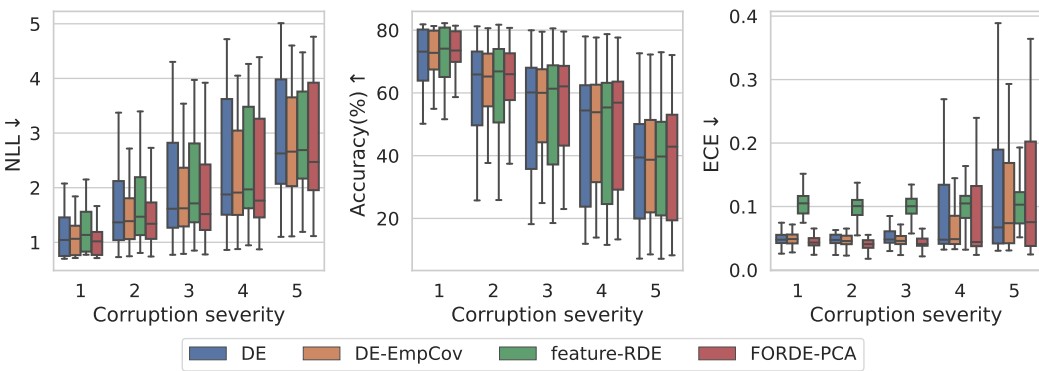

Figure 8: **FoRDE performs better than baselines across all metrics and under all corruption severities.** Results for RESNET18/CIFAR-100-C. Each ensemble has 10 members.

## D.3 COMPARISON BETWEEN FoRDE-PCA AND EMPCOV PRIOR

In Section 3.3, we discussed a possible connection between FoRDE-PCA and the EmpCov prior (Izmailov et al., 2021a). Here, we further visualize performance of FoRDE-PCA, DE with EmpCov prior and vanilla DE on different types of corruptions and levels of severity for the RESNET18/CIFAR10 setting in Fig. 10. This figure also includes results of FoRDE-PCA with EmpCov prior to show that these two approaches can be combined together to further boost corruption robustness of an ensemble. Overall, Fig. 10 shows that FoRDE-PCA and DE-EmpCov have similar behaviors on the majority of the corruption types, meaning that if DE-EmpCov is more or less robust than DE on a corruption type then so does FoRDE-PCA. The exceptions are the *blur* corruption types ({*motion, glass, zoom,*

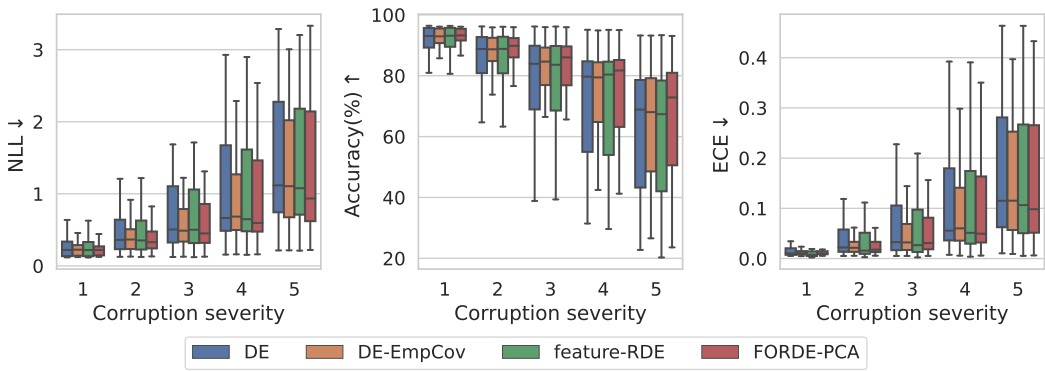

Figure 9: **FoRDE performs better than baselines across all metrics and under all corruption severities.** Results for RESNET18/CIFAR-10-C. Each ensemble has 10 members.

*defocus, gaussian*}-blur), where DE-EmpCov is less robust than vanilla DE while FoRDE-PCA exhibits better robustness than DE. Finally, by combining FoRDE-PCA and EmpCov prior together, we achieve the best robustness on average.

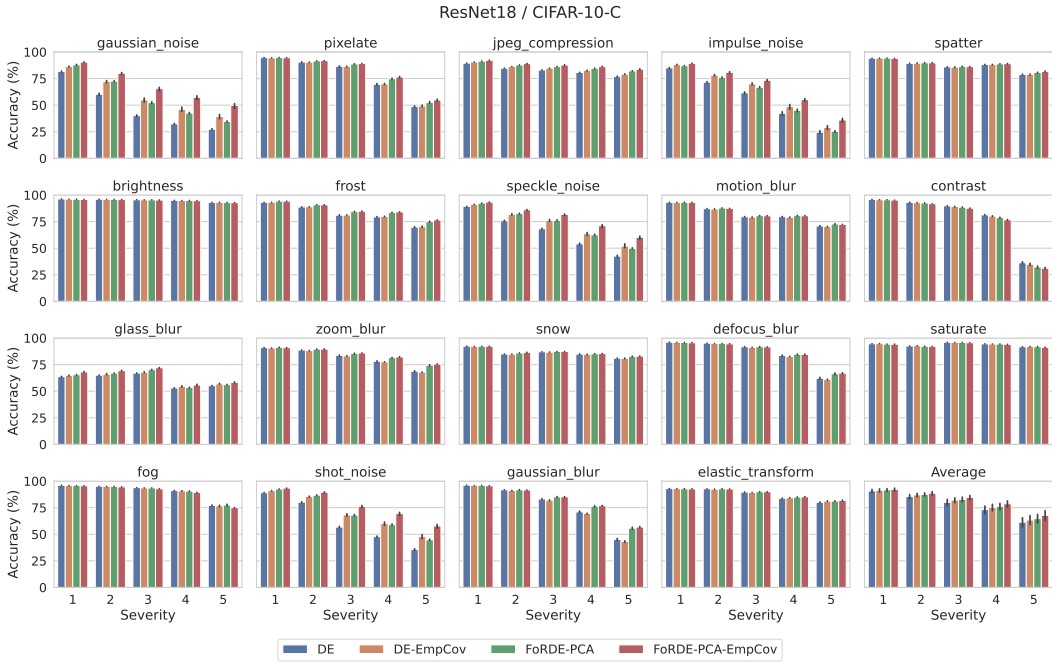

Figure 10: **FoRDE-PCA and EmpCov prior behave similarly in most of the corruption types** Here we visualize accuracy for each of the 19 corruption types in CIFAR-10-C in the first 19 panels, while the last panel (bottom right) shows the average accuracy. Both FoRDE-PCA and DE-EmpCov are more robust than plain DE on most of the corruption types, with the exception of *contrast* where both FoRDE-PCA and DE-EmpCov are less robust than DE. On the other hand, on the *blur* corruption types ({*motion, glass, zoom, defocus, gaussian*}-blur), DE-EmpCov is less robust than vanilla DE while FoRDE-PCA exhibits better robustness than DE.

### D.4 TUNING THE LENGTHSCALES FOR THE RBF KERNEL

In this section, we show how to tune the lengthscales for the RBF kernel by taking the weighted average of the identity lengthscales and the PCA lengthscales introduced in Section 3.3. Particularly,

using the notation of Section 3.3, we define the diagonal lengthscale matrix $\boldsymbol{\Sigma}_\alpha$:

$$\boldsymbol{\Sigma}_\alpha = \alpha\boldsymbol{\Lambda}^{-1} + (1-\alpha)\mathbf{I} \tag{25}$$

where $\boldsymbol{\Lambda}$ is a diagonal matrix containing the eigenvalues from applying PCA on the training data as defined in Section 3.3. We then visualize the accuracy of FoRDE-PCA trained under different $\alpha \in \{0.0, 0.1, 0.2, 0.4, 0.8, 1.0\}$ in Fig. 11 for the RESNET18/CIFAR-100 setting and in Fig. 12 for the RESNET18/CIFAR-10 setting. Fig. 11 shows that indeed we can achieve good performance on both clean and corrupted data by choosing a lengthscale setting somewhere between the identity lengthscales and the PCA lengthscales, which is at $\alpha = 0.4$ in this experiment. A similar phenomenon is observed in Fig. 12, where $\alpha = 0.2$ achieves the best results on both clean and corrupted data.

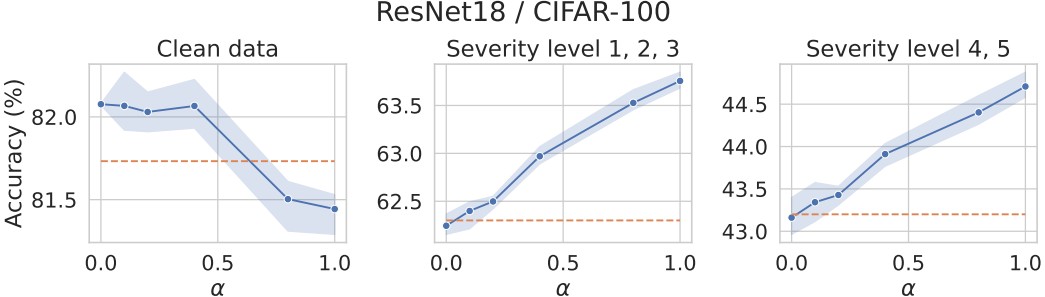

Figure 11: **When moving from the identity lengthscales to the PCA lengthscales, FoRDE becomes more robust against natural corruptions, while exhibiting small performance degradation on clean data.** Results are averaged over 3 seeds. Blue lines show performance of FoRDE, while orange dotted lines indicate the average accuracy of DE for comparison. At the identity lengthscales, FoRDE has higher accuracy than DE on in-distribution data but are slightly less robust against corruptions than DE. As we move from the identity lengthscales to the PCA lengthscales, FoRDE becomes more and more robust against corruptions, while showing a small decrease in in-distribution performance. Here we can see that $\alpha = 0.4$ achieves good balance between in-distribution accuracy and corruption robustness.

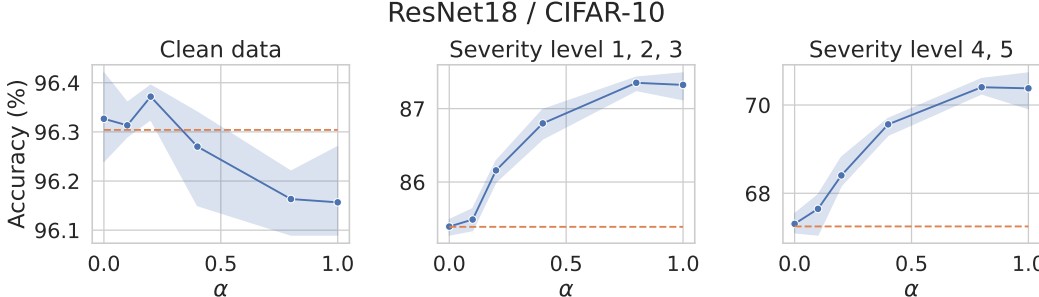

Figure 12: **When moving from the identity lengthscales to the PCA lengthscales, FoRDE becomes more robust against natural corruptions, while exhibiting small performance degradation on clean data.** Results are averaged over 3 seeds. Blue lines show performance of FoRDE, while orange dotted lines indicate the average accuracy of DE for comparison. At the identity lengthscales, FoRDE has higher accuracy than DE on in-distribution data but are slightly less robust against corruptions than DE. As we move from the identity lengthscales to the PCA lengthscales, FoRDE becomes more and more robust against corruptions, while showing a small decrease in in-distribution performance. Here we can see that $\alpha = 0.2$ achieves good balance between in-distribution accuracy and corruption robustness.

