# OpenReview forum: "Input-gradient space particle inference for neural network ensembles"
_ICLR.cc/2024/Conference — ICLR 2024 spotlight_

### Official Review · Reviewer_aXJD · 2023-10-29

**Soundness:** 3 good
**Presentation:** 3 good
**Contribution:** 2 fair
**Rating:** 6
**Confidence:** 4

**Summary:**

Most prior research employing particle-based variational inference (ParVI) has proven to be inefficient and has not significantly improved performance. To tackle these issues, this study presents a new ParVI approach known as the First-order Repulsive Deep Ensemble (FoRDE), which integrates repulsion principles into the realm of first-order input gradients.

**Strengths:**

- The idea of incorporating repulsion into first-order input gradients(not a function space or a weight space repulsion which are quite common in Bayesian Neural Network literature) to enhance functional diversity is new to the community and intriguing.
- The paper is well-written, ensuring it is easy to read and understand.

**Weaknesses:**

- The scale of experiments are quite small to show the effectiveness of FoRDE.
- The overall performance gain compared to other baselines looks quite marginal for the out-of-distribution datasets, especially for the TinyImageNet which is the largest dataset. And shows lower performance compared to the other baselines for the in-distribution datasets.
- Having empirical or theoretical evidence to demonstrate the effectiveness of FoRDE in enhancing input gradient diversity would be beneficial.
- Providing empirical results that illustrate how the improved input gradient diversity effectively changes into enhanced functional diversity in deep neural network scenarios would be valuable.
- Additional hyperparameters for the kernel would be another burden for this method.

**Questions:**

See the weakness section

Recommend
- It is recommended to include an ethics statement and a reproducibility statement right after the main paper.

---

> ### Author Response · Authors · 2023-11-15
>
> Thanks for your review. We have updated our manuscript according to your comments and uploaded the new version to the system. Below we address the specific concerns.
>
> > The scale of experiments are quite small to show the effectiveness of FoRDE.
>
> In our paper, we performed experiments using ResNet18 and PreActResNet18 architectures with CIFAR-10/100 and TinyImageNet datasets. These experiments have been designed to show effectiveness of FoRDE on scales comparable to earlier works on particle inference for neural networks [1, 2, 3] and neural network ensembles [4, 5].
>
> > The overall performance gain compared to other baselines looks quite marginal for the out-of-distribution datasets, especially for the TinyImageNet which is the largest dataset. And shows lower performance compared to the other baselines for the in-distribution datasets.
>
> While our method achieve slightly lower performance on clean data compared to the baselines, we argue that the performance gain on corrupted data is much more substantial and is not trivial to achieve since we need not assume any prior knowledge of these corruptions, and FoRDE consistently outperforms the baselines on corruptions across different datasets. For instance, Table 1 and 2 in the manuscript shows that  FoRDE-PCA achieves a +1.3% gain on CIFAR-100-C and +2.4% gain on CIFAR-10-C in accuracy compared to the second-best results.
>
> > Having empirical or theoretical evidence to demonstrate the effectiveness of FoRDE in enhancing input gradient diversity would be beneficial.
>
> > Providing empirical results that illustrate how the improved input gradient diversity effectively changes into enhanced functional diversity in deep neural network scenarios would be valuable.
>
> To show that FoRDE indeed produces ensembles with higher input gradient diversity among member models, which in turn leads to higher functional diversity than DE, we visualize the input gradient distance and epistemic uncertainty of FoRDE and DE in Fig. 7 of Section D.1 in the Appendix of the revised manuscript. To measure the differences between input gradients, we use *cosine distance*, defined as $1-\cos(\mathbf{u}, \mathbf{v})$ where $\cos(\mathbf{u}, \mathbf{v})$ is the cosine similarity between two vectors $\mathbf{u}$ and $\mathbf{v}$. To quantify functional diversity, we calculate the epistemic uncertainty using the formula in [6], similarly to the transfer learning experiments. Fig. 7 shows that FoRDE has higher gradient distances among members compared to DE, while also having higher epistemic uncertainty across all levels of corruption severity. Intuitively, as the test inputs become more corrupted, epistemic uncertainty of both FoRDE and DE increases, and the input gradients between member models become more dissimilar for both methods. These results suggest that there could be a connection between input gradient diversity and functional diversity in neural network ensembles.
>
> > Additional hyperparameters for the kernel would be another burden for this method.
>
> We did acknowledge this difficulty in our manuscript, and proposed setting the lengthscales based on the PCA of the training data which makes the model more robust against natural corruptions. For the revised version of the manuscript, we also included Section D.4 in the Appendix which shows that by choosing a lengthscale setting which is a linear combination of the identity lengthscales and the PCA lengthscales, FoRDE can achieve good performance on both clean and corrupted data. This can serve as a guideline for choosing the lengthscales when applying FoRDE on new datasets.
>
> > It is recommended to include an ethics statement and a reproducibility statement right after the main paper.
>
> Thank you for your recommendation. We have included the ethics statement and reproducibility statement at the end of the revised manuscript, noting that these sections do not count towards the page limit according to the ICLR author guidelines.
>
> [1] Z. Wang, T. Ren, J. Zhu, and B. Zhang, “Function Space Particle Optimization for Bayesian Neural Networks,” in ICLR 2019.
>
> [2] F. D’Angelo and V. Fortuin, “Repulsive deep ensembles are Bayesian,” In NeuRIPS, 2021.
>
> [3] S. Yashima, T. Suzuki, K. Ishikawa, I. Sato, and R. Kawakami, “Feature Space Particle Inference for Neural Network Ensembles,” in ICML 2022.
>
> [4] A. Rame and M. Cord, “DICE: Diversity in Deep Ensembles via Conditional Redundancy Adversarial Estimation,” in ICLR 2021.
>
> [5] T. Pang, K. Xu, C. Du, N. Chen, and J. Zhu, “Improving adversarial robustness via promoting ensemble diversity,” in ICLR 2019.
>
> [6] S. Depeweg, J.-M. Hernandez-Lobato, F. Doshi-Velez, and S. Udluft, “Decomposition of Uncertainty in Bayesian Deep Learning for Efficient and Risk-sensitive Learning,” in ICML 2018.

---

> > ### Comment · Reviewer_aXJD · 2023-11-18
> > **Response to Authors**
> >
> > I appreciate the authors' comprehensive explanation. While I still have some queries regarding FoRDE's performance, I've adjusted my evaluation positively and now lean towards accepting it.

---

### Official Review · Reviewer_XVK8 · 2023-10-30

**Soundness:** 4 excellent
**Presentation:** 4 excellent
**Contribution:** 3 good
**Rating:** 8
**Confidence:** 4

**Summary:**

This paper is concerned with adapting particle based variational inference for improved training of neural network ensembles.  The authors attempt to circumvent problems that have affected previous attempts to use particle based variational inference for ensembles, with a lack of effective repulsion in weight space (intended to promote functional diversity) chief among them.  This paper proposes instead to enforce diversity in the input gradients rather than in weight space, by using Wasserstein gradient descent along with an RBF kernel defined over the input gradients to guide the particles during training.  They compare against deep ensembles and other BNNs on accuracy, calibration, and robustness to covariate shift.

**Strengths:**

- The paper is very well written.  The potential advantages of moving to input gradient based diversity management are well introduced & well motivated, and the explanations are largely self-contained, which is no small feat considering page restrictions for conferences.
- In particular, the main contribution section (section 3) is *so* well written.  It takes time to lead the reader from the wider view of Wasserstein gradient descent, to input space gradients, and the more narrow questions of choice of kernels, and their tradeoffs.  Of all the papers I reviewed, this was by far the most enjoyable and informative to read.  Bravo for taking the time to write so clearly.

**Weaknesses:**

- One thing I often worry about is that the experiments are performed only in the vision domain, on over-hygenic datasets.  While I don't want to discount the amount of work needed to extend to other domains, projects like [WILDS](https://wilds.stanford.edu/) make this easier, and build confidence that demonstrated success isn't due to some quirk of CIFAR datasets.
- One other complaint that to the authors' credit they highlight in section 3.5 is the cost of computing FoRDEs.  At a 3x computational premium to DEs, the penalty paid in compute seems to be the largest drawback of FoRDE with respect to DEs.  Do the authors have any ideas for reducing this burden? DEs themselves are expensive in both space and time to compute.

**Questions:**

- Regarding the motivation of the RBF kernel in the \textbf{Choosing the base kernel} paragraph of section 3.2, they are good arguments for using the RBF kernel, but are there others that were considered? As the authors suggest in section 3.3 and 3.4, choosing the length scales for RBF presents its own problem.  Could this be circumvented by employing a simpler base kernel?
- Again in section 3.3, is the median heuristic required? I’m a little unsure at the outset why this is the solution chosen over any others that would reduce the effect of the dominant eigenvalues.

---

> ### Author Response · Authors · 2023-11-15
>
> Thank you for the positive review.
>
> > One thing I often worry about is that the experiments are performed only in the vision domain, on over-hygenic datasets. While I don't want to discount the amount of work needed to extend to other domains, projects like WILDS make this easier, and build confidence that demonstrated success isn't due to some quirk of CIFAR datasets.
>
> Thank you for your suggestion. We were not aware of the WILDS project. We will certainly start using this project in our future works for more robust evaluations. Here, we followed previous works on Repulsive deep ensembles [1, 2] which focused mainly on benchmarking on vision datasets. We also provided experiment results on TinyImageNet, which is a bigger dataset with a higher number of classes (200 classes) compared to CIFAR.
>
> > One other complaint that to the authors' credit they highlight in section 3.5 is the cost of computing FoRDEs. At a 3x computational premium to DEs, the penalty paid in compute seems to be the largest drawback of FoRDE with respect to DEs. Do the authors have any ideas for reducing this burden? DEs themselves are expensive in both space and time to compute.
>
> We are currently investigating methods to circumvent this computational drawback of FoRDE. One approach that we are studying is to replace the input-output Jacobian with the Jacobian-vector product. This approach can be implemented in Jax using the `jax.jvp` function, which efficiently calculates the Jacobian-vector product during the forward pass using the dual number approach [3]. In our preliminary experiments, we found that this new approach is almost as fast as plain DEs. We have added this discussion in Section 6 of the revised manuscript.
>
> > Regarding the motivation of the RBF kernel in the \textbf{Choosing the base kernel} paragraph of section 3.2, they are good arguments for using the RBF kernel, but are there others that were considered? As the authors suggest in section 3.3 and 3.4, choosing the length scales for RBF presents its own problem. Could this be circumvented by employing a simpler base kernel?
>
> Here, we employed the RBF kernel following prior works on particle inference for neural networks [1, 2] and we did not consider other kernels for this task. While it is difficult to set the length scales for the RBF kernels, we also demonstrated that these length scales allow us to control the biases of the resulting ensemble, since we can improve the ensemble’s robustness against natural corruptions using the PCA length scale setting. For alternative kernels, we can explore the family of kernels on the unit sphere introduced in [4] since we compare input gradients on a unit sphere, which is an interesting future research direction. Another solution is to study dimensionality reduction techniques for input gradients before calculating the kernel, which also reduces the number of length scales to be set. We have added this discussion in Section 6 of the revised manuscript.
>
> > Again in section 3.3, is the median heuristic required? I’m a little unsure at the outset why this is the solution chosen over any others that would reduce the effect of the dominant eigenvalues.
>
> Since we calculate the kernelized repulsion term using the RBF kernel, we need to use the median heuristic so that the RBF kernel does not vanish during optimization. To understand this, we can look at Eq. (14) in the manuscript. If the eigenvalues are large, then the square distance inside the exponential function of the RBF kernel can reach a large value during optimization, and thus the RBF kernel will quickly converge to 0 and the repulsion term will also reach 0. Therefore, we divide the square distance inside the exponential with a scalar estimated via the median heuristic so that the repulsion term does not vanish during optimization. Prior works [5, 6] also employ the median heuristic to avoid this vanishing problem when using the RBF kernel in particle variational inference.
>
> [1] F. D’Angelo and V. Fortuin, “Repulsive deep ensembles are Bayesian,” In NeuRIPS, 2021.
>
> [2] S. Yashima, T. Suzuki, K. Ishikawa, I. Sato, and R. Kawakami, “Feature Space Particle Inference for Neural Network Ensembles,” in ICML 2022.
>
> [3] J. Bradbury et al., “JAX: composable transformations of Python+NumPy programs.” 2018.
>
> [4] S. Jayasumana, R. Hartley, M. Salzmann, H. Li, and M. Harandi, “Optimizing over radial kernels on compact manifolds,” in CVPR 2014.
>
> [5] Q. Liu and D. Wang, “Stein Variational Gradient Descent: A General Purpose Bayesian Inference Algorithm,” in NeuRIPS, 2016.
>
> [6] C. Liu, J. Zhuo, P. Cheng, R. Zhang, and J. Zhu, “Understanding and Accelerating Particle-Based Variational Inference,” in ICML, 2019.

---

> > ### Comment · Reviewer_XVK8 · 2023-11-15
> > **Response to authors**
> >
> > > Here, we employed the RBF kernel following prior works on particle inference for neural networks [1, 2] and we did not consider other kernels for this task. While it is difficult to set the length scales for the RBF kernels, we also demonstrated that these length scales allow us to control the biases of the resulting ensemble, since we can improve the ensemble’s robustness against natural corruptions using the PCA length scale setting. For alternative kernels, we can explore the family of kernels on the unit sphere introduced in [4] since we compare input gradients on a unit sphere, which is an interesting future research direction.
> >
> > I'm glad to see the authors have already considered how to move from RBF kernels (whose suitability they have established in my mind) to other families with an established way of optimizing the parameters.  I look forward to seeing if they'll make a difference.
> >
> > > Another solution is to study dimensionality reduction techniques for input gradients before calculating the kernel, which also reduces the number of length scales to be set. We have added this discussion in Section 6 of the revised manuscript.
> >
> > I applaud this idea, but am somewhat skeptical that reducing the information in the input gradients by dimensionality reduction will lead to a benefit.  It's worth investigating though.
> >
> > > Since we calculate the kernelized repulsion term using the RBF kernel, we need to use the median heuristic so that the RBF kernel does not vanish during optimization. To understand this, we can look at Eq. (14) in the manuscript. If the eigenvalues are large, then the square distance inside the exponential function of the RBF kernel can reach a large value during optimization, and thus the RBF kernel will quickly converge to 0 and the repulsion term will also reach 0. Therefore, we divide the square distance inside the exponential with a scalar estimated via the median heuristic so that the repulsion term does not vanish during optimization. Prior works [5, 6] also employ the median heuristic to avoid this vanishing problem when using the RBF kernel in particle variational inference
> >
> > Thanks for pointing out the necessity of the median heuristic.  So if I understand your explanation correctly, this is a necessary measure to guard against undesired behaviour in optimization, something akin to why (e.g) gradient clipping is employed?  That makes more sense now.

---

> > > ### Author Response · Authors · 2023-11-15
> > >
> > > > Thanks for pointing out the necessity of the median heuristic. So if I understand your explanation correctly, this is a necessary measure to guard against undesired behaviour in optimization, something akin to why (e.g) gradient clipping is employed? That makes more sense now.
> > >
> > > Yes, you are correct that the median heuristic is employed to avoid undesirable behaviour during optimization.

---

### Official Review · Reviewer_eVm5 · 2023-11-01

**Soundness:** 2 fair
**Presentation:** 3 good
**Contribution:** 2 fair
**Rating:** 6
**Confidence:** 3

**Summary:**

This paper points out that while the repulsion in the existing weight-space or function-space repulsive deep ensembles has been theoretically well-motivated, it does not lead to a practical performance improvement compared to vanilla deep ensembles. Rather than relying on repulsion in weight or function space, the authors employ a kernel comparing input gradients of particles and propose First-order Repulsive Deep Ensembles (FoRDE). Experimental results clearly indicate that FoRDE outperforms baseline methods, particularly when dealing with corrupted data.

**Strengths:**

1. I have experienced that although repulsive deep ensembles are theoretically well-grounded, they do not result in performance enhancements in practice. In this regard, this paper is well-motivated, as it states, "Neither weight nor function space repulsion has led to significant improvements over vanilla DEs."
2. The paper provides a comprehensive overview of the literature concerning repulsive deep ensembles. Also, the proposed approach is meticulously detailed in a step-by-step manner, as well as its practical considerations.
3. The connection to the EmpCov prior (Izmailov et al., 2021) further clarifies why the proposed FoRDE-PCA algorithm performs well for data under common corruptions.

---
Izmailov et al., 2021,  Dangers of Bayesian model averaging under covariate shift.

**Weaknesses:**

Despite the critique that neither weight nor function space repulsion yielded significant improvements compared to vanilla DEs, the FoRDE algorithm introduced in this context still did not result in a substantial performance enhancement over vanilla DEs. In particular, FoRDE-Identity demonstrates a performance similar to that of vanilla DE, while FoRDE-PCA excels in performance under corruption but significantly diminishes its in-distribution performance.

The authors seem to have recognized this aspect; "Hence, we believe that the optimal lengthscales for good performance on both clean and corrupted data lie somewhere between unit lengthscales (the identity) and using the inverse square root eigenvalues as lengthscales." For this paper to be considered complete, it should not just acknowledge such ideal lengthscales but also offer experimental evidence of their practical identification.

**Questions:**

1. The paper mentions the reasons for the ineffectiveness of weight-space repulsion: (1) "Typically repulsion is done in the weight space to capture different regions in the weight posterior. However, due to the over-parameterization of neural networks, weight-space repulsion suffers from redundancy." (2) "Weight-space repulsion is ineffective due to difficulties in comparing extremely high-dimensional weight vectors and the existence of weight symmetries (Fort et al., 2019; Entezari et al., 2022)." Could you provide a more detailed explanation of this?

2. The paper outlines the advantages of ensemble methods in four specific areas: (1) predictive performance, (2) uncertainty estimation, (3) robustness to adversarial attacks, and (4) corruptions. In the experimental results, it delves into (1) using ACC, (2) using NLL and ECE, and (4) using cA, cNLL, and cECE. Did you carry out any experiments regarding (3) by any chance? Considering that the current experimental results are somewhat lacking in (1) and (2), it might be worthwhile to focus more on (3) and (4).

3. FoRDE-PCA exhibits robust performance in addressing common corruptions (although it shows a minor decrease in its in-distribution performance). Hence, I would like to suggest providing more detailed experimental results concerning common corruptions, e.g., if it operates similarly to EmpCov (Izmailov et al., 2021), it is worth exploring whether the most beneficial corruption type aligns as well.

---
Fort et al., 2019, Deep ensembles: A loss landscape perspective.
Entezari et al., 2022, The role of permutation invariance in linear mode connectivity of neural networks.
Izmailov et al., 2021,  Dangers of Bayesian model averaging under covariate shift.

---

> ### Author Response · Authors · 2023-11-15
> **Response (1/2)**
>
> Thank you for the reviews. We have made changes to our manuscript based on your suggestions and uploaded the new version to the system. Below we address the specific concerns.
>
> >  For this paper to be considered complete, it should not just acknowledge such ideal lengthscales but also offer experimental evidence of their practical identification.
>
> We have included in Section D.4 in the Appendix of the revised manuscript anecdotal evidence that the ideal lengthscale setting lies somewhere between the identity lengthscales and the PCA lengthscales for the ResNet18 / CIFAR-100 experiments. Particularly, we train FoRDE under the lengthscale settings $\alpha * \mathrm{pca\\_lengthscales} + (1-\alpha) * \mathrm{identity\\_lengthscale}$, where we increase $\alpha$ from $0$ to $1$, and visualize the accuracy on both clean and corrupted data as a function of $\alpha$ in Fig. 11 in the Appendix.
> Fig. 11 shows that as $\alpha$ increases, FoRDE becomes more robust against corruptions, while exhibiting small degradation in performance on clean data. With $\alpha \in [0.1, 0.4]$, FoRDE achieves higher accuracy than DE on both clean and corrupted data, with $\alpha=0.4$ produces the best result.
>
> > The paper mentions the reasons for the ineffectiveness of weight-space repulsion: (1) "Typically repulsion is done in the weight space to capture different regions in the weight posterior. However, due to the over-parameterization of neural networks, weight-space repulsion suffers from **redundancy**." (2) "Weight-space repulsion is ineffective due to difficulties in comparing extremely high-dimensional weight vectors and the existence of **weight symmetries** (Fort et al., 2019; Entezari et al., 2022)." Could you provide a more detailed explanation of this?
>
> Performance of an ensemble partly relies on the functional diversity of its members. The purpose of weight-space repulsion in ensemble learning is to find weight particles that are different from each other in the hope that they represent different functions in the function space. However, the weight posterior of a neural network contains weight symmetries, meaning that two different weight vectors can represent the same function [1]. If we average the predictions from these two weight vectors, it would be equivalent to using the prediction from one of those weight vectors, and we refer to this problem as **redundancy**. Additionally, modern neural networks contains large amounts of parameters (more than tens of millions of parameters) arranged into hierarchical structures with various components (such as non-linear activations, convolution, normalization and pooling layers), making the geometry of the weight posterior highly complex [2]. This complexity makes it difficult to define a meaningful kernel to measure the similarity between two weight vectors, which is needed to calculate the kernelized repulsion term. Previous works [3, 4] used the RBF kernel for weight-space repulsion and observed no improvement in performance compared to plain Deep ensembles.
>
>
> [1] C. M. Bishop, Pattern recognition and machine learning. Chapter 5.1. 2006
>
> [2] R. Entezari, H. Sedghi, O. Saukh, and B. Neyshabur, “The Role of Permutation Invariance in Linear Mode Connectivity of Neural Networks,” in ICLR, 2022.
>
> [3] F. D’Angelo and V. Fortuin, “Repulsive deep ensembles are Bayesian,” In NeuRIPS, 2021.
>
> [4] S. Yashima, T. Suzuki, K. Ishikawa, I. Sato, and R. Kawakami, “Feature Space Particle Inference for Neural Network Ensembles,” in ICML 2022.

---

> > ### Author Response · Authors · 2023-11-15
> > **Response (2/2)**
> >
> > >  The paper outlines the advantages of ensemble methods in four specific areas: (1) predictive performance, (2) uncertainty estimation, (3) robustness to adversarial attacks, and (4) corruptions. In the experimental results, it delves into (1) using ACC, (2) using NLL and ECE, and (4) using cA, cNLL, and cECE. Did you carry out any experiments regarding (3) by any chance? Considering that the current experimental results are somewhat lacking in (1) and (2), it might be worthwhile to focus more on (3) and (4).
> >
> > We would like to start by pointing out that the purpose of our work is to study input gradient repulsion as a viable way to improve functional diversity of ensembles, and thus we performed experiments to confirm that it does improve functional diversity (through visualization on small experiments in Figs. 2 and 3 and through the transfer learning experiments in Section 5.4).  Following prior works on repulsive ensembles [1, 2], we evaluated FoRDE on natural image corruptions, and studied how to choose the hyperparameters that allow FoRDE to be more robust against corruptions than the baselines. While these hyperparameters do slightly decrease performance of FoRDE on clean data, we believe that the improvement in performance on natural image corruptions is much more substantial and is not trivial to achieve since FoRDE has no prior knowledge of these corruptions. For instance, Table 1 and 2 in the manuscript shows that FoRDE-PCA achieves a +1.3% gain on CIFAR-100-C and +2.4% gain on CIFAR-10-C in accuracy compared to the second-best results. Thus, we did not perform any experiments or analysis on robustness of FoRDE to adversarial attacks.
> >
> > While we agree that further evaluation of FoRDE on adversarial robustness would enhance our paper, we think it would require a separate in-depth study to analyze how FoRDE can be modified to achieve robustness against adversarial attacks, and how it fares against other methods specifically designed for this task. Including this study in our current manuscript would make it much longer than a typical conference paper.
> >
> > > FoRDE-PCA exhibits robust performance in addressing common corruptions (although it shows a minor decrease in its in-distribution performance). Hence, I would like to suggest providing more detailed experimental results concerning common corruptions, e.g., if it operates similarly to EmpCov (Izmailov et al., 2021), it is worth exploring whether the most beneficial corruption type aligns as well.
> >
> > Thank you for the suggestion and we have provided more detailed experimental results comparing FoRDE-PCA to the EmpCov prior in Section D.3 in the Appendix of the revised manuscript. Specifically, we visualize the accuracy of DE, DE-EmpCov, FoRDE-PCA and FoRDE-PCA-EmpCov on each corruption type in CIFAR-10-C for the ResNet18/CIFAR-10 experiments in Fig. 10, where FoRDE-PCA-EmpCov means that we trained FoRDE-PCA with the EmpCov prior. Fig. 10 shows that both FoRDE-PCA and EmpCov operate similarly on the majority of the corruption types, meaning that if DE-EmpCov is more or less robust than DE on a corruption type then so is FoRDE-PCA. The exceptions are the *blur* corruption types (*{motion, glass, zoom, defocus, gaussian}*-blur), where DE-EmpCov is less robust than vanilla DE while FoRDE-PCA exhibits better robustness than DE. Finally, by combining FoRDE-PCA and EmpCov prior together, we achieve the best robustness on average.
> >
> > [1] F. D’Angelo and V. Fortuin, “Repulsive deep ensembles are Bayesian,” In NeuRIPS, 2021.
> >
> > [2] S. Yashima, T. Suzuki, K. Ishikawa, I. Sato, and R. Kawakami, “Feature Space Particle Inference for Neural Network Ensembles,” in ICML 2022.

---

> > > ### Comment · Reviewer_eVm5 · 2023-11-18
> > > **Response to authors**
> > >
> > > I appreciate the authors' efforts in addressing my concerns. I anticipate the revised paper will comprehensively address the mentioned issues in the main text (e.g., it would be nice to see FoRDE with tuned alpha values in the main tables). I am increasing the score since the additional results have resolved my primary concerns.
> > >
> > > Moreover, I would like to make some remarks on the transfer learning experiments outlined in Section 5.4, employing a 3-hidden layer MLP after the fixed pre-trained feature extractor. Considering the relatively small size of this model, it would be feasible to conduct posterior sampling using the gold standard sampling method, Hamiltonian Monte Carlo (HMC). Introducing an HMC baseline, in my opinion, would offer readers additional valuable insights.

---

> > > > ### Author Response · Authors · 2023-11-19
> > > >
> > > > > I appreciate the authors' efforts in addressing my concerns. I anticipate the revised paper will comprehensively address the mentioned issues in the main text (e.g., it would be nice to see FoRDE with tuned alpha values in the main tables). I am increasing the score since the additional results have resolved my primary concerns.
> > > >
> > > > > Moreover, I would like to make some remarks on the transfer learning experiments outlined in Section 5.4, employing a 3-hidden layer MLP after the fixed pre-trained feature extractor. Considering the relatively small size of this model, it would be feasible to conduct posterior sampling using the gold standard sampling method, Hamiltonian Monte Carlo (HMC). Introducing an HMC baseline, in my opinion, would offer readers additional valuable insights.
> > > >
> > > > Thank you for increasing the score and for the suggestions that will undoubtedly improve the quality of our work. We will certainly incorporate all the suggestions in the final version of our manuscript. For the current version of the manuscript which is available in the system, we have included the results of FoRDE with tuned lengthscales in the main tables (Tables 1 and 2) and mentioned these results in Section 5.2 in the main text.

---

### Official Review · Reviewer_VQHQ · 2023-11-03

**Soundness:** 3 good
**Presentation:** 3 good
**Contribution:** 4 excellent
**Rating:** 8
**Confidence:** 3

**Summary:**

The paper proposes a novel method for ensembling deep models that ensures diversity of the ensemble members. The paper continues the line of work in particle-based variational inference transforming the repulsion step of this approach into an input gradient space. This is different from the existing works that have done this step in weight and function spaces.

**Strengths:**

* A novel method for an important problem of ensembling
* Thorough empirical evaluation and comparison to the existing methods
* Drawing connections with the existing methods
* The paper is mostly well written and easy to follow
* Runtime analysis presented

**Weaknesses:**

* Some presentation unclearness (see details below)
* Some transformations between theory in Section 3 and steps in Algorithm (in Appendix) are not obvious


1. What corruption is considered? CIFAR-10/100-C datasets have several types of corruptions each of which has several level of severity of corruptions. No confidence intervals (+-) for corruption results.
2. Section 3.1 doesn't address that the target distribution \pi is not available, or am I missing something?
3. It would help to clear some confusion of how Algorithm comes in place if steps in Algorithm would be linked to equations in Section 3.
4. Section 3.4. "However, in practice we found no performance degradation nor convergence issues in our experiments" - though the convergence issues can easily be observed, in order to see no performance degradation one would need to compare the performance with and without mini-batches. This experiment is not presented in the paper (including Appendix).
5. Though the code is provided, some implementation details in text are missing. For example, ECE computation details such as a number of bins. Or details of OOD experiments: what portion of OOD data (CIFAR-100 for CIFAR-100 and vice versa) was used.
6. No reference for CINIC10 dataset

**Questions:**

What exact corruption has been used in reported corruption experiments?

---

> ### Author Response · Authors · 2023-11-15
>
> Thank you for the suggestions which helped us further improve the manuscript, and we have uploaded the new version to the system. Below we address the specific concerns.
>
> > What corruption is considered? CIFAR-10/100-C datasets have several types of corruptions each of which has several level of severity of corruptions. No confidence intervals (+-) for corruption results.
>
> We used all the available types of corruptions and all 5 levels of severity in CIFAR-10/100 and TinyImageNet datasets for evaluation. We have added these details in Section C.2 and C.3 in the Appendix of the updated manuscript. We also added figures containing the corruption results with confidence intervals for each level of severity in Section D.2 in the updated manuscript.
>
> > Section 3.1 doesn't address that the target distribution \pi is not available, or am I missing something?
>
> You are correct. We have clarified that the target distribution $\pi$ is intractable (and hence approximated with the particle distribution in Eq. 6)  in Section 3.1 of the revised manuscript.
>
> > It would help to clear some confusion of how Algorithm comes in place if steps in Algorithm would be linked to equations in Section 3.
> We have added clarifications in Section C.1 in the Appendix of the revised manuscript.
> > Section 3.4. "However, in practice we found no performance degradation nor convergence issues in our experiments" - though the convergence issues can easily be observed, in order to see no performance degradation one would need to compare the performance with and without mini-batches. This experiment is not presented in the paper (including Appendix).
>
> We wholeheartedly agree with this assessment. We however cannot perform the full-batch experiments due to resource limitations, since performing forward-backward passes on the entire 50000 training samples of CIFAR requires a large amount of GPU memory. One could compromise by varying the minibatch size, however it has been observed that large-batch training produces models with lower generalization performance than small-batch training [1], meaning that we would likely observe lower performance on large batch sizes compared to small batch sizes due to this phenomenon. Therefore, we have changed this statement to “However, in practice we found no convergence issues in our experiments'' in Section 3.4 of the revised manuscript.
>
> > Though the code is provided, some implementation details in text are missing. For example, ECE computation details such as a number of bins. Or details of OOD experiments: what portion of OOD data (CIFAR-100 for CIFAR-100 and vice versa) was used.
>
> For all experiments, we used 15 bins to calculate ECE. For the OOD experiments, we calculated the epistemic uncertainty on the test sets of CIFAR-10/100 and CINIC10. We have added these details in Section C.2 and C.3 in the Appendix of the updated manuscript.
>
> > No reference for CINIC10 dataset.
>
> We have added the reference in our updated manuscript.
>
> [1] N. S. Keskar, D. Mudigere, J. Nocedal, M. Smelyanskiy, and P. T. P. Tang, “On Large-Batch Training for Deep Learning: Generalization Gap and Sharp Minima,” in ICLR 2017.

---

> > ### Comment · Reviewer_VQHQ · 2023-11-18
> >
> > Thank you so much for addressing all my comments/questions.
> >
> > Could you please elaborate about corruptions more please? How do you compute 1 number for each of the metrics (cA, cNLL, cECE) based on all corruptions and all severity types? Do you just consider all corruptions and all severity types as one big dataset?
> >
> > Thank you again, I enjoyed your paper (as it seems other reviewers as well), hopefully it will be accepted.

---

> > > ### Author Response · Authors · 2023-11-18
> > >
> > > Thank you for enjoying our work.
> > >
> > > > Could you please elaborate about corruptions more please? How do you compute 1 number for each of the metrics (cA, cNLL, cECE) based on all corruptions and all severity types? Do you just consider all corruptions and all severity types as one big dataset?
> > >
> > > For each of the corruption metrics (cA, cNLL and cECE), we consider all corruption types and all severity levels as one big test dataset, as has been done in [1, 2].
> > >
> > > [1] M. Dusenberry et al., “Efficient and Scalable Bayesian Neural Nets with Rank-1 Factors,” in ICML 2020.
> > >
> > > [2] S. Yashima, T. Suzuki, K. Ishikawa, I. Sato, and R. Kawakami, “Feature Space Particle Inference for Neural Network Ensembles,” in ICML 2022.

---

> > > > ### Comment · Reviewer_VQHQ · 2023-11-20
> > > >
> > > > Thank you so much for clarification. It is just different from, e.g., Ovadia, Y., Fertig, E., Ren, J., Nado, Z., Sculley, D., Nowozin, S., Dillon, J., Lakshminarayanan, B. and Snoek, J., 2019. Can you trust your model's uncertainty? evaluating predictive uncertainty under dataset shift. Advances in neural information processing systems, 32., that is why I was confused. Could you please add this clarification to the manuscript?

---

> ### Author Response · Authors · 2023-11-20
>
> We have added clarification in the current version of the manuscript available on the system. Specifically, we added the following sentence in the first paragraph of Section 5.2 in the main text:
>
> > For evaluations on input perturbations, we use CIFAR-10/100-C and TINYIMAGENET-C provided by Hendrycks & Gimpel (2017), which are datasets of corrupted test images containing 19 image corruption types across 5 levels of severity, and we report the accuracy, NLL and ECE averaged over all corruption types and severity levels (denoted cA, cNLL and cECE in Tables 1–4).

---

### Meta-Review · Area_Chair_gr8J · 2023-12-06

**Metareview:**

This paper proposes a novel version of repulsive deep ensembles, where the repulsion is computed on the gradients with respect to the input. It argues that these gradients correspond to the features learnt by the respective ensemble members and shows empirically that this approach leads to improved uncertainty estimation. After an active discussion between authors and reviewers, all reviewers lean towards accepting this paper (two of them strongly so). While the reviewers praised the motivation, the originality of the idea, the treatment of the related work, and the thorough empirical evaluation, they were critical of the clarity of the presentation, the runtime of the approach, a lack of ablation studies, and the experimental results. However, most (if not all) of these issues have been addressed in the extensive rebuttal. It therefore seems warranted to accept the paper, with the understanding that the authors will continue their efforts to fully address the reviewer feedback in the camera-ready version.

**Justification For Why Not Higher Score:**

It is unclear whether this topic is interesting for a wide enough subset of the ICLR audience to warrant an oral

**Justification For Why Not Lower Score:**

The reviewers were all quite excited about the paper, also in the post-rebuttal discussion

---

### Decision · Program_Chairs · 2024-01-16

Accept (spotlight)